# Achieving a solar-to-chemical efficiency of 3.6% in ambient conditions by inhibiting interlayer charges transport

Yuyan Huang[1], Minhui Shen[1], Huijie Yan[2], Yingge He[3], Jianqiao Xu[1], Fang Zhu[1], Xin Yang [3], Yu-Xin Ye [2,4] ✉ & Gangfeng Ouyang [1,2,4] ✉

Efficiently converting solar energy into chemical energy remains a formidable challenge in artificial photosynthetic systems. To date, rarely has an artificial photosynthetic system operating in the open air surpassed the highest solar-to-biomass conversion efficiency (1%) observed in plants. In this study, we present a three-dimension polymeric photocatalyst achieving a solar-to-$H_2O_2$ conversion efficiency of 3.6% under ambient conditions, including real water, open air, and room temperature. The impressive performance is attributed to the efficient storage of electrons inside materials via expeditious intramolecular charge transfer, and the fast extraction of the stored electrons by $O_2$ that can diffuse into the internal pores of the self-supporting three-dimensional material. This construction strategy suppresses the interlayer transfer of excitons, polarizers and carriers, effectively increases the utilization of internal excitons to 82%. This breakthrough provides a perspective to substantially enhance photocatalytic performance and bear substantial implications for sustainable energy generation and environmental remediation.

Solar-to-chemical conversion (SCC) provides a promising avenue for resolving the energy and environmental crises that afflict contemporary society by harnessing the largest renewable energy sources on Earth[1–8]. Among the diverse artificial photocatalytic systems currently available, the photosynthesis of hydrogen peroxide ($H_2O_2$) offers a compelling strategy, enabling cost-effective synthesis in an open system and maximizing photon efficiency through simultaneous utilization of oxidation and reduction reactions.

Photocatalytic processes generally involve several steps. Firstly, the catalyst can be excited to a depth of up to 100 nanometers by photons to generate excitons, which then transform into loosely bound polarons. Typically, the internal excitons and polarons need to migrate to the surface and further separate into free charges to react with the substrates[9]. Whereas, the transfer distance of excitons or polarons in organic photocatalysts is usually no more than 20 nanometers, and a massive amount of them recombine before reaching the surface[1,10,11]. Despite the successful separation of polarons into free charges before recombination, the internal free charges still require to traverse the interlayer to reach the surface, which recombine inevitably when the electrons and holes from different layers get close under the Coulomb attraction force. It has been reported that >90% of excitons recombine rapidly within sub-microsecond (sub-μs), leading to unpromising photocatalytic efficiency of $H_2O_2$ which is far from the requirements of practical applications[12].

To overcome the recombination of excitons, polarons or charge carriers during the migration, one of the most studied methods is to reduce the transfer distance by constructing ultrathin two-dimensional (2D) semiconductors (Fig. 1)[13]. However, these low-dimensional

[1]Key Laboratory of Bioinorganic and Synthetic Chemistry of Ministry of Education, LIFM, School of Chemistry, IGCME, Sun Yat-sen University, Guangzhou 510275, China. [2]School of Chemical Engineering and Technology, IGCME, Sun Yat-sen University, Zhuhai 519082, China. [3]School of Environmental Science and Engineering, Guangdong Provincial Key Laboratory of Environmental Pollution Control and Remediation Technology, Sun Yat-sen University, Guangzhou 510275, China. [4]Southern Marine Science and Engineering Guangdong Laboratory (Zhuhai), Zhuhai, Guangdong 519082, China. ✉e-mail: yeyuxin5@sysu.edu.cn; cesoygf@mail.sysu.edu.cn

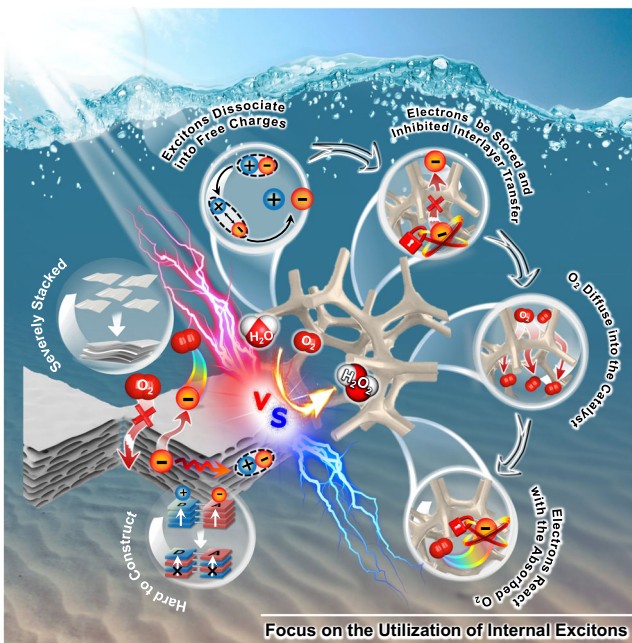

**Fig. 1 | Schematic illustration comparing previous works and this work.** In this work, we propose a strategy to enhance the utilization of internal excitons by suppressing the recombination of metastable excitons, polarons, and photo-generated charge carriers during interlayer transfer.

materials are always severely stacked owing to their high surface energies, resulting in bulk recombination of free charges during transportation. Another plausible solution to this challenge is to construct electron donor-acceptor (D-A) polymeric photocatalysts possessing ordered donor-on-donor (for hole transport) and acceptor-on-acceptor (for electron transport) bi-continuous π-columnar structures were proposed[14–18]. Unfortunately, it is challenging to precisely construct such kind of highly ordered microstructures, and donor-on-acceptor alternative stacking is more likely to form owing to favorable electrostatic interactions, which tends to induce the recombination of electrons and holes[19]. Additionally, the severely stacked structures also embed the dominant active sites inside the photocatalysts and only leave small portions of the active sites on the outer surfaces. Thus, the optimized utilization of internal excitons and active sites is a feasible means to develop high-performance photocatalysts.

Herein, we propose a strategy to enhance the utilization of internal excitons by suppressing the recombination of metastable excitons, polarons, and photogenerated charge carriers during interlayer transfer (Fig. 1). This is achieved by inhibiting their interlayer transfer through rapid dissociation of polarons into free charges via expeditious intramolecular electron transfer, followed by their storage within the catalyst. Furthermore, by exposing storage sites through the three-dimensional architecture of the photocatalyst, reactants can effectively access and extract free or stored charge carriers from deep within the material, preventing recombination during interlayer transfer. This approach yields a break-through photocatalytic rate of $H_2O_2$ ranging from 9257 to 9991 $\mu mol \cdot g^{-1} \cdot h^{-1}$, accompanied by a solar-to-chemical conversion efficiency reaching 3.6% under ambient conditions, i.e. real water, open air and room temperature. To our best knowledge, this is an impressive SCC achieved among the photo-synthetic systems of $H_2O_2$, and is among the rare instances where SCC in ambient conditions has surpassed the highest solar-to-biomass conversion (SBC) rate of typical plants. Importantly, this study introduces an approach for attributing peaks in femtosecond transient absorption spectroscopy, which is challenging and prone to confusion. By utilizing sacrificial agents and small molecule monomers, we

differentiate and identify various photoinduced transient species (excitons, polarons, electrons). Based on this approach, the mechanism of accelerated electron extraction from the three-dimensional catalysts on photophysical processes was further explored by switching atmospheres in situ. These techniques enable the clarification of the transfer pathways (interlayer or intramolecular) and provide insights into the corresponding proportions of each transfer method. Specifically, most of the free electrons (82.2% in air and 89.2% in $O_2$) were utilized through intramolecular transfer, while the long-distance interlayer electron transport was substantially suppressed, thus circumvented the recombination during interlayer transport. This study offers a perspective to substantially enhance photocatalysis performance by improving photon utilization and bears substantial implications for sustainable energy generation and environmental remediation.

## Results
### Synthesis and characterization of catalysts
The self-supporting three-dimensional (3D) amorphous photocatalyst was synthesized with triptycenes (TPC) as the self-supporting electron donors, the built-in redox anthraquinone (AQ) moieties as the electron acceptors, and alkynyl as connectors (Fig. 2a). This photocatalyst was named as TPC-3D. The three-dimensional structure of triptycenes units exposes more active sites and allows $O_2$ to diffuse into the interior of the material. Alkynyl bridges have superior electron transfer properties due to the linear conjugated structure. Furthermore, AQ moieties possess a strong electron withdrawal capacity, which promotes the separation of photo-induced carriers. Critically, AQ has a two-electron storage capacity which effectively inhibits the recombination of charge carriers[2]. This 3D photocatalyst can rapidly store electrons on AQ by intramolecular transfer and allow $O_2$ diffuse into the catalyst interior to extract electrons, which suppressing the interlayer transport of photogenerated excitons, polarons, and charge carriers. For comparison, another two two-dimensional (2D) structural analogs were synthesized by using pyrene (PYR) and triphenylene (TPL) as the planar electron donors, which were termed PYR-2D and TPL-2D, respectively (Fig. 2a). The method for introducing the alkyne group was shown in Supplementary Fig. 1, and the three conjugated polymers (CPs) were conventionally synthesized through a one-step Sonogashira reaction. The determination of electron donor and acceptor relies on the distribution positions of their Highest Occupied Molecular Orbital (HOMO) and Lowest Unoccupied Molecular Orbital (LUMO) in their optimized structures. As depicted in Supplementary Fig. 2, HOMO is predominantly located in TPC, while LUMO is primarily distributed in the AQ moieties.

The characteristic signals of alkynyl groups were observed in solid-state cross-polarization/magic-angel-spinning (CP/MAS) $^{13}C$ NMR spectra (Supplementary Fig. 3), Raman spectra (Supplementary Fig. 4), and in FT-IR spectra (Supplementary Fig. 5). Besides, the FT-IR spectra showed the stretching bands of C=O at -1673 cm$^{-1}$ (Supplementary Fig. 5), combined with the characteristic benzoquinone groups (-181 ppm) in $^{13}C$ NMR spectra (Supplementary Fig. 3) and X-ray photoelectron spectroscopy (XPS) measurements (Supplementary Figs. 6 and 7), manifesting the retaining of AQ in CPs[20]. More details are reported in the supplementary information to demonstrate the successful polymerization for all the CPs through the Sonogashira reaction (Supplementary Figs. 2–7)[21]. The powder X-ray diffraction (PXRD) profiles revealed that all the CPs exhibited the features of amorphous carbon[22] (Supplementary Fig. 8). Scanning electron microscopy (SEM) and transmission electron microscopy (TEM) showed that the CPs possessed rough surfaces and uniform structures (Supplementary Figs. 9 and 10). Also, as shown in Fig. 2b, all the CPs exhibited strong absorption throughout the visible light region and even in the near-infrared region, which demonstrated that the CPs possessed large conjugation structures. In TPC-3D, the three phenylene rings of the

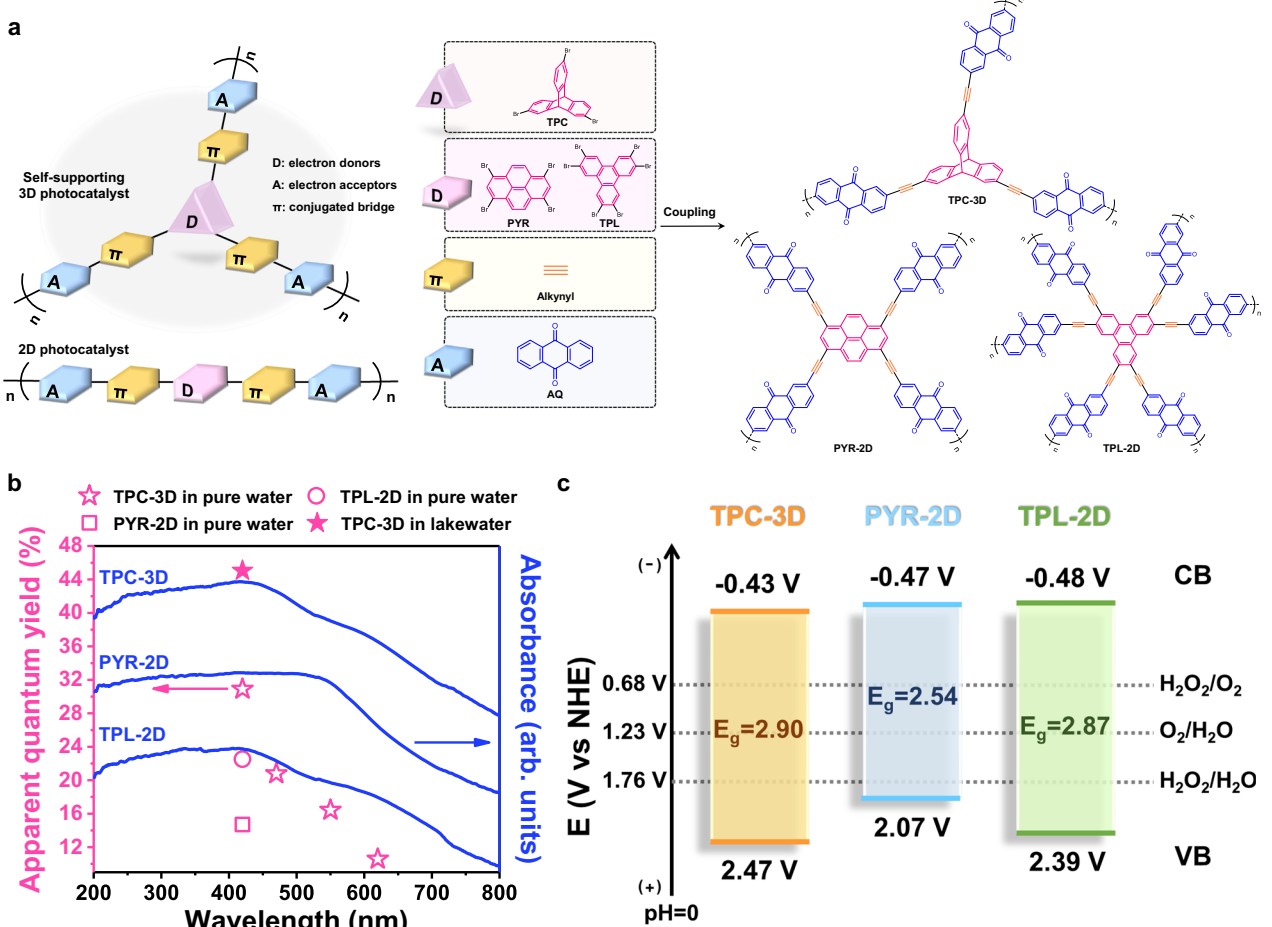

**Fig. 2 | Structural characteristics of catalysts. a** Chemical components of catalysts: D, electron donors; A, electron acceptors; π, conjugated bridge. **b** UV-visible diffuse reflectance spectra of the catalysts, along with apparent quantum yield (AQY) at specified wavelengths. The CPs demonstrated strong absorption in the visible and even near-infrared regions, indicating their large conjugation structures. **c** Schematic illustration of the electronic band structures of the catalysts.

triptycene influence each other electronically through space (which is often called homoconjugation), which also extends its conjugation structure[23]. In addition, all CPs exhibited satisfactory thermal stability (Supplementary Fig. 11).

The energy band structures of the CPs were subsequently determined (Fig. 2c). The conduction band (CB) minima of TPC-3D, PYR-2D, and TPL-2D were determined to be −0.43 eV, −0.47 eV, and −0.48 eV versus the normal hydrogen electrode (NHE) via the Mott-Schottky tests (Supplementary Fig. 12)[24–26]. Subsequently, the valence band (VB) maxima was determined using the valence band-XPS (VB-XPS) to measure the energy difference of the VB to Fermi level ($E_{VB-XPS}$) (Supplementary Fig. 13). Then, Kelvin Probe Force Microscopy (KPFM) was used to measure the work function (φ) of the material, representing the potential difference from the Fermi level to the vacuum level (Supplementary Fig. 14)[27]. The VB maxima referenced to the normal hydrogen electrode ($E_{NHE}$) could be obtained as 2.47 eV, 2.07 eV, and 2.39 eV for TPC-3D, PYR-2D, and TPL-2D (Fig. 2c and Supplementary Note 1). Thus, it could be concluded that all CPs were capable for the 2 e⁻ oxygen reduction reaction (ORR) and the 2 e⁻ or 4 e⁻ water oxidation reaction (WOR)[25].

## Photocatalytic performance

The photosynthesis of $H_2O_2$ by the CPs was evaluated under simulated sunlight (xenon-lamp light, >400 nm, 100 mW·cm⁻²). Pure water and air atmosphere were adopted with no sacrificial agents or continuous $O_2$ supplied. TPC-3D exhibited the highest $H_2O_2$ production rate

among the CPs, i.e. 5940 μmol·g⁻¹·h⁻¹, which was 2.4 and 1.8 times that of PYR-2D and TPL-2D (Fig. 3a). Moreover, in real water including lake water, river water, and seawater, the photosynthetic rates of TPC-3D were even further improved to break-through values as high as 9991 μmol·g⁻¹·h⁻¹, 9615 μmol·g⁻¹·h⁻¹, and 9257 μmol·g⁻¹·h⁻¹, respectively (Fig. 3a). These efficiencies were much higher than most reported for reactions conducted in air, pure $O_2$, or even with the use of hole sacrificial agents (Fig. 3b and Supplementary Table 1)[2,4,20,28–38]. In addition, TPC-3D essentially maintained its high efficiency, morphology, and component for 5 cycles (Supplementary Figs. 15–18 and Supplementary Note 2). To assess the potential for practical application, TPC-3D was coated onto a glass slide and immersed it in real water for photocatalytic reactions (Fig. 3c). Remarkably, the catalyst was without appreciable loss in performance even after 15 cycles (Fig. 3d), and relevant structural characterizations have provided (Supplementary Fig. 19 and Supplementary Note 2).

Besides, the apparent quantum yield (AQY) of TPC-3D reached up to 30.9%, 20.8%, and 10.6% at 420, 470, and 620 nm in pure water, respectively, and even as high as 44.8% at 420 nm and 11.9% at 620 nm in lake water (Fig. 2b). The AQYs at all the wavelengths are among the leading values reported for the photosynthesis of $H_2O_2$ (Supplementary Table 2). Significantly, the solar-to-chemical energy conversion (SCC) process achieved successful implementation under low concentration of photocatalyst usage (0.4 g·L⁻¹) and in open air conditions (Supplementary Fig. 20, Supplementary Movies 1 and 2), with an impressive efficiency of 2.4% in pure water and 3.6% in lake water

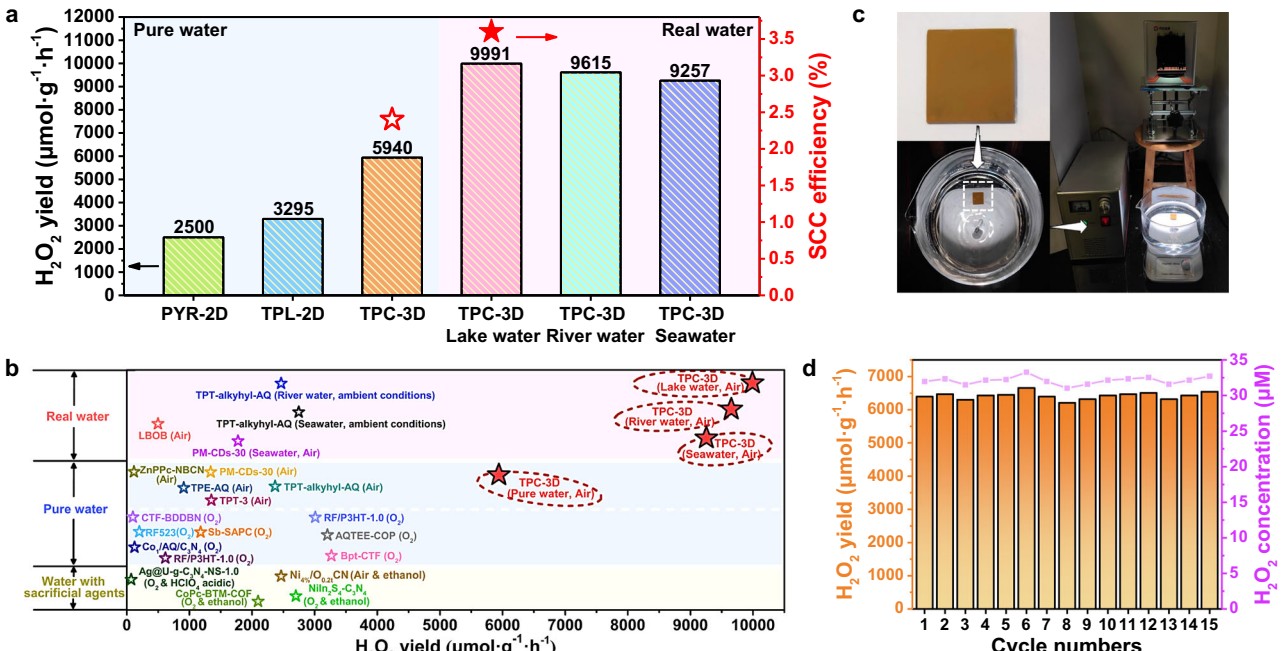

**Fig. 3 | Photocatalytic performance. a** Photocatalytic production of $H_2O_2$ by CPs in open air under different water conditions. Experimental conditions: $\lambda > 400$ nm (xenon lamp, light intensity: 100 mW·cm$^{-2}$), photocatalyst (1 mg), water samples (50 mL). **b** Comparison of the photocatalytic performance for $H_2O_2$ between TPC-3D and other reported photocatalysts under various water samples[2,4,20,28–38]. **c** The photograph of photocatalytic reaction of TPC-3D in real water. Photocatalytic reaction setup with: a glass slide (20 mm × 20 mm) loaded with 2 mg of TPC-3D powders (top left), placed in 400 mL of lake water (bottom left), and exposed to a xenon lamp ($\lambda > 400$ nm, light intensity: 100 mW·cm$^{-2}$). **d** The stable $H_2O_2$ production activity of TPC-3D under the condition described in Fig. 3c.

(Fig. 3a). These SCC efficiencies were superior to most of those reported for photosynthetic systems of $H_2O_2$ (Supplementary Table 3) and were even higher than the highest solar-to-biomass conversion efficiency of typical plants (1%)[20,29]. It is notable that other artificial photocatalytic reactions, such as water splitting and carbon dioxide reduction, cannot be performed in the open air. The current photocatalytic system provides a more feasible way for SCC.

In addition, it was not foreseen that the photosynthetic rate of $H_2O_2$ by TPC-3D was increased by >55% in real water. Generally, the photocatalytic rate would be decreased in real water due to the intricate interferential components, i.e. inorganic ions, and organic matters[20,39–41]. It was also observed that the increasing rate was positively correlated with the content of dissolved organic matters (DOMs) in real water and unrelated to the concentration of inorganic carbon, inorganic salts, and pH (Supplementary Table 4). It was inferred that the DOMs served as electron donors for the photosynthesis of $H_2O_2$, since the photocatalytic performance of TPC-3D was similarly increased after adding isopropanol, sodium oxalate, or edetate disodium (EDTA-2Na) as electron donors (Supplementary Fig. 21). Furthermore, it has been reported that DOMs in real water can provide protons[38] or act as electron donors[42–44], thereby promoting photocatalytic performance. Thus, we adopted theoretical calculations to elucidate the sources of protons and electrons in this system. The composition of DOMs was complex, and to simplify the model, phenol was used as an example. As illustrated in Supplementary Fig. 22a, upon light excitation, the electron donor generated an electron and transferred to AQ, where it reacted with the C=O double bond in AQ to form AQH. At this point, an additional electron and a proton were required to form anthrahydroquinone (AQH$_2$)[20]. In the presence of phenol, there were four possibilities for proton and electron sources in this step: $H_2O$ provided protons and electrons, phenol provided protons and electrons, water provided protons while phenol provided electrons, and phenol provided protons while $H_2O$ provided electrons. The reaction energies ($\Delta E$) of these four pathways were obtained by

theoretical calculation as 2.30 eV, 0.89 eV, 3.55 eV and 5.58 eV respectively (Supplementary Fig. 22b), indicating that AQH preferred proton-coupled electron transfer through phenol rather than $H_2O$ to form AQH$_2$. Furthermore, phenol was added into pure water for photocatalytic experiments and the photocatalytic performance was improved significantly (Supplementary Fig. 23). Thus, these results substantiate that organic pollutants can supply electrons and protons, facilitating the photocatalytic production of hydrogen peroxide. Moreover, TPC-3D demonstrated exceptional resistance to ionic interference (Supplementary Fig. 23 and Supplementary Note 3). The even higher photosynthetic performance of TPC-3D in real water can open up an application of TPC-3D in real water remediation via in situ photosynthesis of $H_2O_2$.

### Photochemical process

To reveal the underlying reasons for the enhanced performance of TPC-3D, we first identified the generation pathway of $H_2O_2$ in CPs and its corresponding active sites involved in the photochemical process. According to the energy band structures of CPs, it could be concluded that all CPs were effective for the oxidation of water into $H_2O_2$ or $O_2$ via the two- or four-electron pathways (Fig. 2c)[25]. The products of the WOR process were confirmed to be $H_2O_2$ and $O_2$ by the Rotating Ring-Disk Electrodes (RRDE) measurements (Supplementary Fig. 24). The time-dependent density functional theory (TD-DFT) calculations indicated that the active sites were mainly located on the TPC and alkynyl moieties for WOR, as the holes were primarily occupied in these two sites in the excites states (Fig. 4a and Supplementary Fig. 25). Specifically, for TPC-3D, WOR occurred at carbon 3 (on triptycenes) and carbon 37 (on alkynyl group) (Fig. 4b). The calculation results showed that it was much favored to adsorb OH* intermediates on alkynyl (Fig. 4c), enabling $H_2O$ to form the target product $H_2O_2$ through a two-electron WOR. The same phenomenon was observed in PYR-2D and TPL-3D (Supplementary Figs. 26, 27). Additionally, the in situ diffuse reflectance infrared Fourier transform spectroscopy (DRIFTS) under $H_2O$

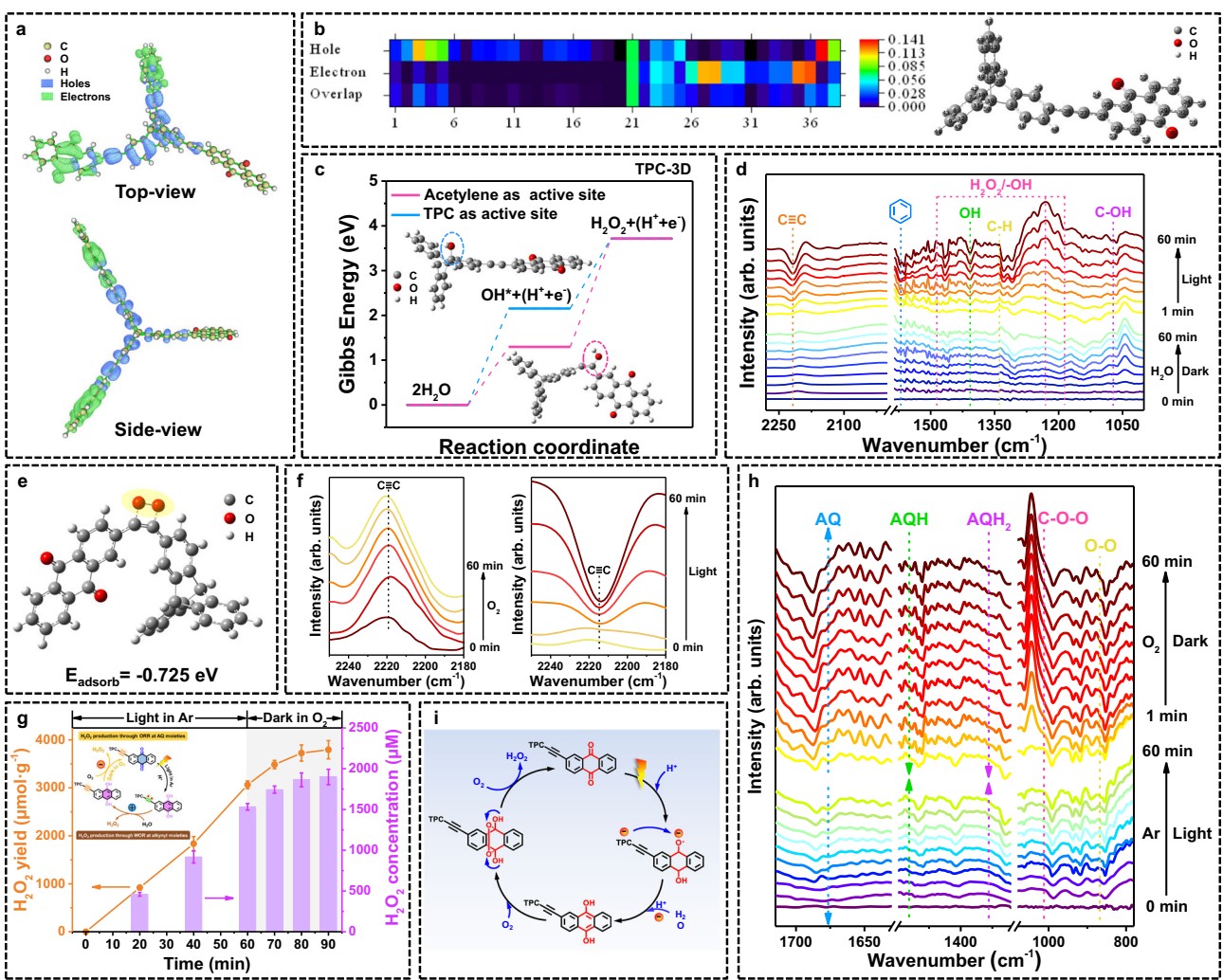

**Fig. 4 | Active sites and pathways for H₂O₂ generation. a** Distribution of holes (blue) and electrons (green) in TPC-3D obtained through the time-dependent density functional theory (TD-DFT) calculations (Isosurface value = 0.001). **b** The contribution of non-hydrogen atoms to holes and electrons in excited state and the corresponding atom labels. **c** Calculated free energy diagrams of two-electron water oxidation pathways toward H₂O₂ production on acetylene and electron-donor sites in TPC-3D. Inset: the adsorption configuration of OH* on different sites of TPC-3D. **d** In-situ DRIFTS under H₂O of TPC-3D. **e** Molecular model and adsorption energy (E_adsorb) for TPC-3D when oxygen is adsorbed at alkynyl moieties. **f** In-situ DRIFTS of TPC-3D adsorbing O₂ in the dark and then irradiating. **g** H₂O₂ yields through ORR at AQ moieties. Inset: the mechanism underlying the experimental processes. Photosynthesis was conducted in an argon atmosphere for 1 h, followed by immediate injection of pure O₂ into the photocatalytic system after stopping the illumination. **h** In-situ DRIFTS of light in argon and dark in oxygen process of TPC-3D. **i** Key steps of H₂O₂ production by two-electron ORR process on AQ.

conditions displayed relatively obvious infrared absorption peaks at 2219, 1573, and 1072 cm⁻¹, which were attributed to C≡C, benzene ring, and C-OH absorption, respectively[45–47] (Fig. 4d). And new peaks emerged around 1184–1203 cm⁻¹, corresponding to the infrared absorption of -OH from generated H₂O₂. These results confirmed the active participation of alkynyl and triptycenes structures in the production of H₂O₂ through the WOR. Moreover, under bubbling argon to remove O₂ from the air, and with the incorporation of electron sacrificial agents (AgNO₃) to inhibit the generation of H₂O₂ from ORR, the formation of H₂O₂ could also be observed, demonstrating that H₂O₂ can be generated via WOR pathway (Supplementary Fig. 28). Isotope experiments also provided strong evidence for WOR (Supplementary Fig. 29).

On the other hand, the synthesis of H₂O₂ was significantly suppressed under conditions of argon and the addition of AgNO₃ (Supplementary Fig. 28), indicating that the photocatalytic generation of H₂O₂ primarily proceeds via ORR. For ORR, all CPs exhibited effective two-electron oxygen reduction to produce H₂O₂ (Supplementary Fig. 30). DFT calculation revealed that O₂ could spontaneously adsorb

on the alkynyl moieties to form the endoperoxide species in TPC-3D (Fig. 4e and Supplementary Fig. 31). To further explore the adsorption behavior of O₂ on alkynyl moieties, in-situ DRIFTS was conducted. As shown in Fig. 4f, the stretching vibration of the C≡C bond (2219 cm⁻¹) increased gradually with the injection of O₂ in the dark. The results indicated that alkynyl groups served as the active site for O₂ adsorption as the adsorbed intermediate could result in increased force constants due to symmetry breaking[45]. In contrast, under irradiation, the signals of alkynyl moieties decreased, suggesting that alkynyl participates in the photocatalytic ORR. This portion of O₂ adsorbed on the alkynyl moieties was proven capable of directly reacting with e⁻ and being conversed to H₂O₂, since H₂O₂ could still be successfully produced under the condition of in Ar atmosphere to exclude O₂ from the air and adding hole sacrificial agents added to inhibit the generation of O₂ from water oxidation (Supplementary Fig. 32).

Meanwhile, another oxygen reduction pathway that reduced AQ reacted with O₂ was also proposed. The TD-DFT calculations showed that the active sites were mainly located on the carbonyl oxygen (atoms 35 and 36 in Fig. 4b) and carbonyl carbon (atoms 27 and 28 in

Fig. 4b) of AQ moieties for ORR. And the superior electrons accepting ability of the AQ moieties was also verified from the LUMO diagrams as well as from the electron paramagnetic resonance (EPR) spectra (Supplementary Figs. 2 and 33). To further confirm the reaction between reduced AQ and $O_2$, the photosynthesis was conducted in an argon atmosphere for one hour, and then pure $O_2$ was injected into the photocatalytic system immediately after stopping the illumination. In the absence of light, where electron-hole pairs cannot be generated, the production of $H_2O_2$ in a dark environment solely relies on stored electron (Fig. 4g). Consequently, the generation of $H_2O_2$ persisted for duration of 20 min in dark. These phenomena indicated that the electrons were stored in AQ to reduce $O_2$ to generate $H_2O_2$ for 20 min[29]. In addition, significant color change during the photocatalytic process can be regarded as another visual indicator of the accumulation of electrons. Upon irradiation in an air atmosphere, the material transitions from a brown to an orange-yellow shade, whereas in an argon atmosphere, the material further shifts to a vivid yellow color, which is attributed to the accumulation of electrons (Supplementary Fig. 34). And in the UV/Vis spectra (Supplementary Fig. 34), the increase of absorbance indicated the formation of the reduced species[48]. Moreover, in-situ DRIFTS provided a strong evidence for the formation of reduced species. As shown in Fig. 4h, the peaks at 1358, 1482 and 1680 $cm^{-1}$ were assigned to the one-electron accumulation state of AQ (AQH), two-electron accumulation state of AQ (AQH$_2$) and AQ, respectively[20]. With the progression of irradiation, the peak of AQ diminished in argon, along with the emergence of peaks corresponding to the reduced species AQH and AQH$_2$. Subsequently, upon the removal of the light source and the injection of oxygen, the peaks of AQH and AQH$_2$ decreased while the peak of AQ increased, and the infrared vibration peak assigned to the 1,4-endoperoxide intermediate species at 910 $cm^{-1}$ were significantly enhanced, indicating that the electron-storage reduced species reacted with $O_2$. These phenomena strongly corroborated the electron storage capability of AQ and the occurrence of the ORR on AQ. Consequently, the ORR process on AQ can be summarized as follows (Fig. 4i): under visible-light irradiation, AQ sequentially formed the one-electron accumulation state of AQ (AQH) and two-electron accumulation state of AQ (AQH$_2$) via an electron-coupled hydrogenation reaction[20,33]. Then, AQH$_2$ reacted with $O_2$ to generate AQH$_2$−1,4-endoperoxide, which subsequently coupled with the adjacent hydrogen in the hydroxyl group to release $H_2O_2$ and regenerated the AQ redox center[49]. Also, the in-situ EPR spectra had revealed the electron signal on AQ during the reaction (Supplementary Fig. 35 and Supplementary Note 4).

As summarized above, photo-induced electrons can be directly captured by absorbed $O_2$ in TPC-3D or stored in AQ for subsequent ORR. To elucidate the ORR pathway, EPR spectra were conducted to detect $\cdot O_2^-$ which involved in 2e$^-$ ORR with 5,5-dimethyl−1-pyrroline N-oxide (DMPO) as the spin-trap agent. As depicted in Supplementary Fig. 36a, TPC-3D containing both alkynyl and AQ exhibited the typical six-line characteristic peaks of DMPO·$\cdot O_2^-$, indicative of the presence of superoxide. For comparison, we synthesized a polymer devoid of alkynyl groups using triptycenes as the electron donor and anthraquinone as the electron acceptor, designated as TPC-AQ (Supplementary Fig. 19a). It is noteworthy that TPC-AQ, which contains AQ but not alkynyl, did not exhibit the peak of DMPO·$\cdot O_2^-$ (Supplementary Fig. 36b). The comparison suggested that ORR pathway occurred on the alkynyl via a two-step 2e$^-$ process involving the formation of the $\cdot O_2^-$ intermediates, while the ORR pathway on AQ reduced $O_2$ to produce $H_2O_2$ via a one-step 2e$^-$ process bypassing $\cdot O_2^-$. To investigate the percentage of ORR occurring on different sites, we conducted a photocatalysis experiment with the addition of superoxide dismutase (SOD), which acts as a scavenger of $\cdot O_2^-$. The presence of SOD did not significantly affect the $H_2O_2$ yield of TPC-AQ, while it reduced the yield in TPC-3D by about 19% (Supplementary Fig. 36c), indicating the percentage of ORR pathway on alkynyl involving $\cdot O_2^-$ intermediates less

than one-fifth. And this differential response to SOD treatment confirms our hypothesis regarding the distinct ORR pathways on the alkynyl and AQ moieties. These two pathways coexist, and which pathway occurs depends on whether oxygen is pre-adsorbed or not (See Supplementary Note 5 for details).The investigation revealed that the production pathways of $H_2O_2$ by 2D CPs were analogous (Supplementary Figs. 30, 33, 37, 38). In 2D CPs, it also included the pathway that the AQ stored the photo-generated electron to reduce the charge carrier recombination through back electron transfer effectively[29]. However, the recombination caused by interlayer electron transfer remained unavoidable. The electrons may become trapped inside the catalyst through AQ, preventing them from coming into contact with the oxygen on the surface of catalyst. Alternatively, the electrons may still need to migrate a long distance from the interior of the catalyst to the interface. Both of these processes resulted in a decrease in the electron utilization efficiency. This limitation was successfully addressed in TPC-3D, which boasts a 3D structure that facilitates $O_2$ diffusion into the pores and exposes a multitude of active sites, promoting efficient electron utilization. As shown in Supplementary Fig. 39, TPC-3D possessed a specific surface area 40%–50% larger than that of PYR-2D and TPL-2D due to its 3D structure, allowing better exposure of the active sites. Besides, as seen from the $O_2$ physical adsorption experiments (Supplementary Fig. 40a), $O_2$ temperature programmed desorption ($O_2$-TPD) experiments (Supplementary Fig. 40b), EPR measurements (Supplementary Fig. 33) and $H_2O$ adsorption experiments (Supplementary Fig. 41), TPC-3D possessed more active sites for $O_2$ adsorption, electron storage and WOR than PYR-2D and TPL-2D (see Supplementary Note 6 for details). Interestingly, although the 2D catalysts owned higher ratios of $O_2$ adsorption sites, specifically alkynyl moieties (electron donors:alkynyl = 1:3 in TPC-3D, while it was 1:4 in PYR-2D and 1:6 in TPL-2D), their overall oxygen adsorption capacity was lower. This was owing to the stacking structures of PYR-2D and TPL-2D embedded substantial portions of active sites inside the 2D CPs. In addition, a significant delay in $O_2$ desorption for TPC-3D also indicated that $O_2$ was likely to be adsorbed inside TPC-3D (Supplementary Fig. 40b). The more accessible active sites in TPC-3D were convinced to circumvent the long-distance electron transfer to the surface of 2D materials. Moreover, the photocurrent densities in pure $O_2$ were significantly decreased to only half of that in air in TPC-3D (Fig. 5). It was inferred that the photoinduced electrons were effectively captured by the internal $O_2$ rather than transported to the electrode. On the contrary, the trends of photocurrent densities for PYR-2D and TPL-2D were 3- and 10-fold higher in $O_2$ than that in air. These results demonstrated the photoinduced electrons in the 2D CPs preferred to transport to the electrode. The electrochemical impedance spectra (EIS) also showed that the electrical resistance of TPC-3D was significantly higher than PYR-2D and TPL-2D, indicating that the charges transportation in the two-dimensional catalysts were more fluent (Supplementary Fig. 42).

## Electron utilization

After identifying the photochemical pathways, it is known that there are three pathways for photoinduced electrons to be utilized (Fig. 6). In order to further investigate the underlying mechanism of affecting the three pathways resulting from the increased accessibility of oxygen inside the catalyst due to the 3D structure, we characterized the rates and proportions of electron transfer in air and $O_2$ atmosphere using transient absorption (TA) spectroscopy (Fig. 7 and Supplementary Figs. 43–45)[50,51]. A 400 nm laser was selected to excite the electron donors and the broad absorption ranging from 800 nm to 1200 nm were observed in the TA spectroscopy of CPs. This broad adsorption was assigned to electronic absorption because they decreased in the presence of AgNO$_3$ as electron scavengers[50].

To clarify the reaction times corresponding to these three pathways for photoinduced electrons to be utilized, the TA decay kinetics

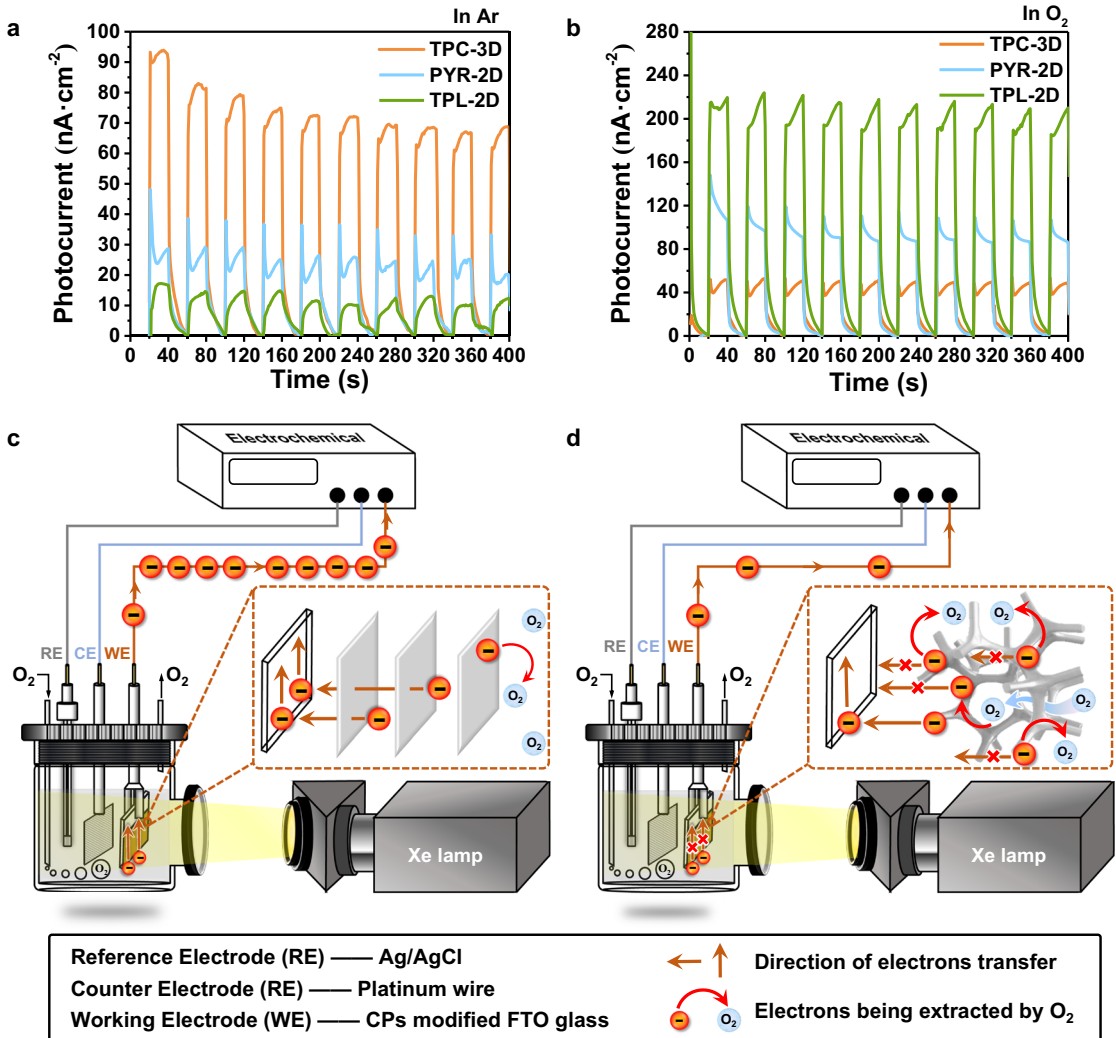

**Fig. 5 | Electron transportation investigation of the photocatalysts.**
**a** Photocurrent measurement curves in 0.1 M Na$_2$SO$_4$ solution under Ar gas bubbling. **b** Photocurrent measurement curves in 0.1 M Na$_2$SO$_4$ solution under O$_2$ gas bubbling. **c** Illustration of the electron transfer mechanism in PYR-2D and TPL-2D with the two-dimensional structure under O$_2$ atmosphere. **d** Illustration of the electron transfer mechanism in TPC-3D with the self-supporting three-dimensional structure under O$_2$ atmosphere.

under O$_2$ and air conditions were fitted by the tri-exponential function, resulting in $\tau_1$, $\tau_2$, $\tau_3$ and their corresponding ratios. In O$_2$, the $\tau_1$ component was accelerated and the proportion was increased, indicating $\tau_1$ could be attributed to the direct reaction between electrons and oxygen (Fig. 6 Pathway I). In comparison, the proportion of $\tau_2$ component was nearly the same in O$_2$ and air conditions, which was likely to be affiliated to the storage of electrons in reduced AQ (Fig. 6 Pathway II). In addition, the $\tau_3$ component was much longer than the other two, which was assigned to electrons transferring across stacked layers to the surfaces and reacting with O$_2$ (Fig. 6 Pathway III). Notably, the contribution of the $\tau_3$ component was found to decrease significantly from 17.8% in the presence of air to 10.8% in the presence of O$_2$ for TPC-3D (Fig. 7d). In contrast, no significant changes were observed in the $\tau_3$ components for PYR-2D and TPL-2D when exposed to O$_2$ or air. This can be attributed to the fact that, in TPC-3D, O$_2$ can penetrate into the interior of the catalyst and reduce the interlayer electron transfer, which ultimately decreases the lifetime of the electrons. Conversely, in two-dimensional materials, O$_2$ cannot penetrate the material's interior, and thus, it does not affect the electron transfer pathway within the catalyst, resulting in no decrease in electron lifetime.

In addition, considering the catalytic reaction occurred under air conditions in an open system, the differences in the electron transfer

pathways of the three CPs in air condition may warrant further exploration. As shown in Fig. 7d, $\tau_3$ of TPC-3D was 55% and 40% shorter than those of PYR-2D and TPL-2D in air, respectively. These results verified that electrons in the bulk of 2D CPs inevitably required long-length transport to the surface for being utilized, whereas free electrons in TPC-3D, they could be rapidly extracted by O$_2$ even in the air condition, thanks to its 3D structure. Thus most of the free electrons of TPC-3D (82.2% in air and 89.2% in O$_2$) were utilized through intramolecular transfer (Pathway I and Pathway II). Remarkably, compared to interlayer transfer (Pathway III, $\tau_3$), intramolecular transfer (Pathway I, $\tau_1$ and Pathway II, $\tau_2$) has a rate that is more than one order of magnitude faster. The estimated lifetimes of the electrons in TPC-3D were 0.90 ps for $\tau_1$, 15.92 ps for $\tau_2$ and 206.60 for $\tau_3$, respectively. Due to the high proportion and faster rate of intramolecular electron transfer in TPC-3D, its estimated electron lifetime was 191.61 ps, which was 50% and 40% lower than PYR-2D and TPL-2D, respectively. Based on the above research, it has been demonstrated that increasing the proportions of intramolecular electron transfer can significantly enhance the total electron transfer rate. Moving forward, the expeditious electron transfer rate effectively improved the electron utilization efficiency. The photocurrent densities in the presence of O$_2$ strongly support the high-efficiency utilization of electrons in TPC-3D, as they

## 2D photocatalyst

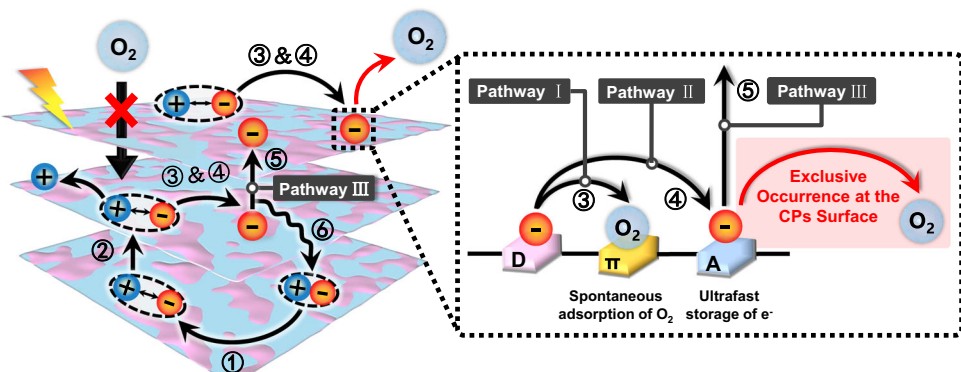

## Self-supporting 3D photocatalyst

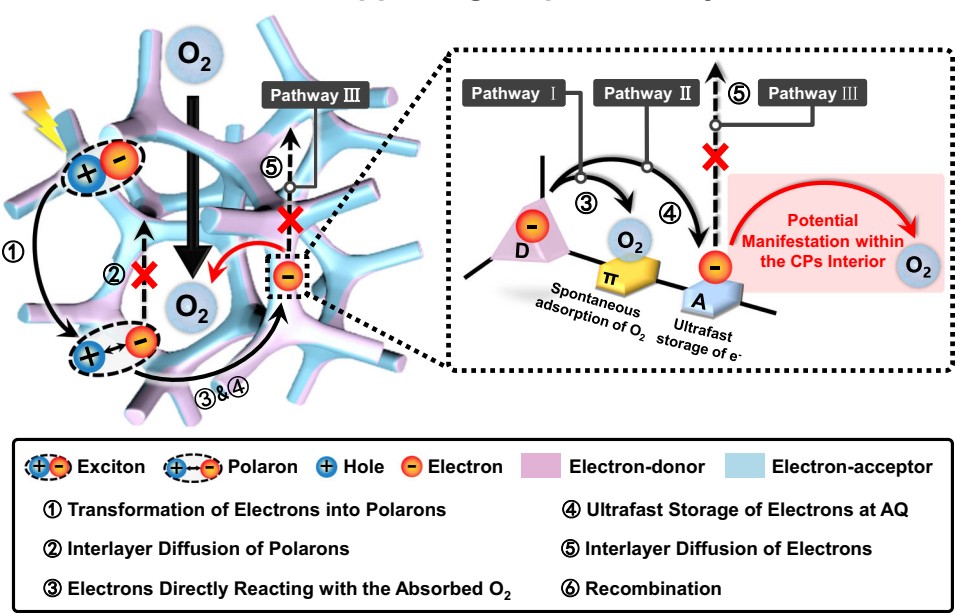

**Fig. 6 | Schematic illustration of the behaviors of excitons, polarons and charge carriers in 2D and 3D photocatalysts.** The behaviors of excitons, polarons and charge carriers are different in 2D and 3D photocatalysts.

were significantly lower than those observed under argon conditions (Fig. 5). This finding demonstrated that the photoinduced electrons were more likely to be extracted by the $O_2$ within the catalyst, rather than being transported to the surface. The improved performance of photocatalytic $H_2O_2$ synthesis in the presence of $O_2$, compared to air, indicated that increasing the accessibility could improve the utilization efficiency of electrons (Supplementary Fig. 46).

In summary, TPC-3D achieved expeditious intramolecular transfer to store electrons in the catalyst and exposed the storage sites through the 3D structure, allowing electrons deep within the material to be efficiently extracted by $O_2$, preventing recombination during interlayer transfer. More importantly, the intramolecular electron transfer ratio increased in TPC-3D, leading to an accelerated total electron transfer rate and improved electronic utilization efficiency (Fig. 8). This provides a promising approach for enhancing the efficiency of photocatalytic systems.

### From polarons dissociation to electron generation

After confirming the faster and more efficient utilization of electrons in TPC-3D, we proceeded to further investigate the impact of this accelerated electron utilization on the polarons dissociation. Firstly, it was necessary to identify the peaks of excitons and polarons in the TA

spectra initially and subsequently measured their respective lifetimes. All the electron donor and CPs exhibited positive peaks immediately and decay rapidly, which were attributed to the absorption of the excitons (Supplementary Figs. 47a, 48a and 49a). The decay of the peaks was attributed to the transformation of the excitons to polarons, pairs of more loosely bound charges by Coulomb attraction forces[52,53]. Subsequently, signals formed after the dissociation of the excitons appeared as the highest positive photoinduced absorption (PIA) peaks at 645 nm in TPC-3D, 720 nm in PYR-2D, and 580 nm in TPL-2D (Supplementary Figs. 47b, 48b, 49b). These signals were drastically reduced regardless of the addition of hole sacrificial agents (sodium oxalate) or electron sacrificial agents ($AgNO_3$) (Fig. 9a and Supplementary Figs. 50–52). Considering that polarons involve a bound state of electrons and holes, it can be inferred that the acceleration of electron or hole extraction through the use of sacrificial agents may potentially expedite the separation of polarons. Thus, these signals could be attributed to polarons. The time delay between the appearance of excitons and polarons signals was the time at which the excitons started to dissociate into polarons, i.e. 630 fs for TPC-3D, 1820 fs for PYR-2D, and 755 fs for TPL-2D (Fig. 9c and Supplementary Figs. 48c, 49c). Owing to the high dissociation efficiencies of the excitons in all the CPs (discussed in detail below), the decay of the PIA peaks was

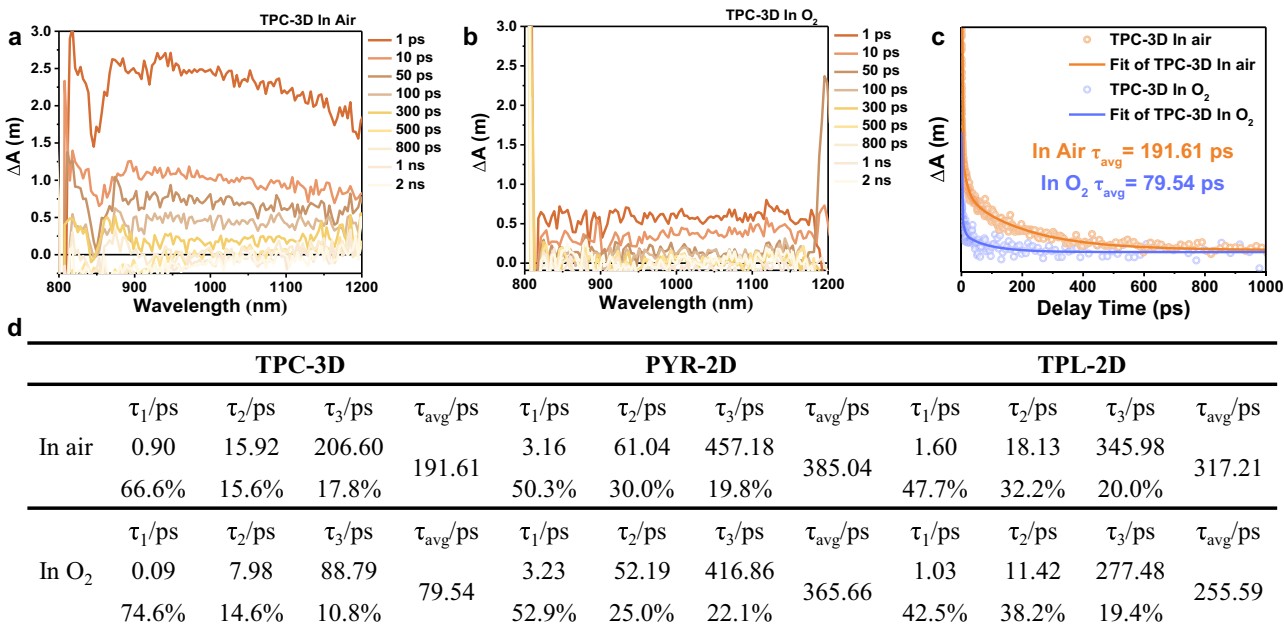

**Fig. 7 | Femtosecond time-resolved transient absorption (fs-TA) measurements in the near-infrared region of TPC-3D. a**, **b** Femtosecond TA spectra in the 800–1200 nm range of TPC-3D under (**a**) air and (**b**) O₂ atmosphere. **c** Comparison of TA kinetic profiles at 1000 nm under air and O₂ conditions, respectively. **d** Decay lifetimes under O₂ and air conditions fitted by the tri-exponential function from corresponding TA kinetic traces at 1000 nm.

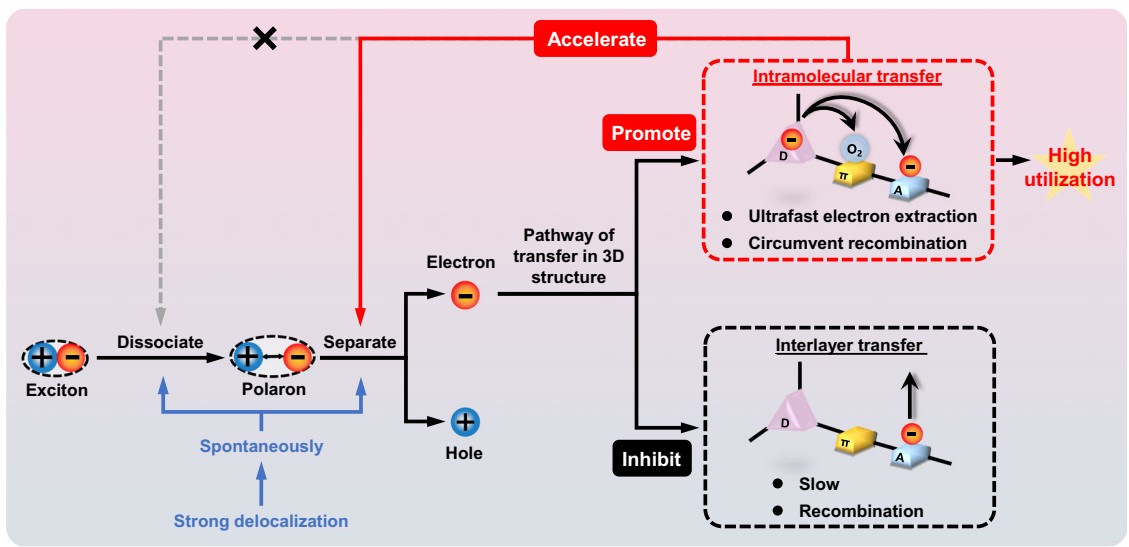

**Fig. 8 | Schematic illustration depicting the interaction among the behavior of excitons, polarons, and electrons in TPC-3D.** There are specific interaction mechanisms in exciton dissociation, polaron transformation, and electron transfer pathways.

assigned to the transformation of polarons into free charges, other than the geminate recombination. The average lifetimes of polarons for TPC-3D, PYR-2D, and TPL-2D were 389.20 ps, 729.54 ps, and 550.26 ps, respectively (Fig. 9d and Supplementary Figs. 51e, 52e). It was notable that the dissociation speed of the polarons for TPC-3D was accelerated by 47% and 29% compared to PYR-2D and TPL-2D.

To further explore the mechanism leading to the accelerated dissociation of the polarons in TPC-3D, the dissociation kinetics of the polarons were fitted with the tri-exponential decay function as shown in Fig. 9b, d, e. The fast components τ₁ and τ₂ were assigned to the dissociation of the polarons formed at the intramolecular D-A interfaces[54,55]. The long-lived component τ₃ was commonly assigned to the dissociation of polarons after interlayer diffusion (Fig. 6 Pathway 2). As shown in Fig. 9e, the ratios of each lifetime component were nearly the same in all the CPs in pure O₂ compared to those in air, which indicated the decay pathways of polarons were not affected by a higher concentration of O₂. However, the average lifetimes ($\tau_{avg}$) and the lifetimes of τ₁, τ₂, τ₃ in O₂ were reduced by 40%, 81%, 46%, and 41% in TPC-3D, respectively, compared with those in air. These results demonstrated that pure O₂ could accelerate the dissociation of polarons. In comparison, each component was changed by <20% for PYR-2D and TPL-2D and all the dissociation pathways of polarons in TPC-3D were faster than that in the 2D CPs (Fig. 9e).

Based on the discussions above, the 3D structure did not alter the pathways of polaron transfer. However, it significantly accelerated

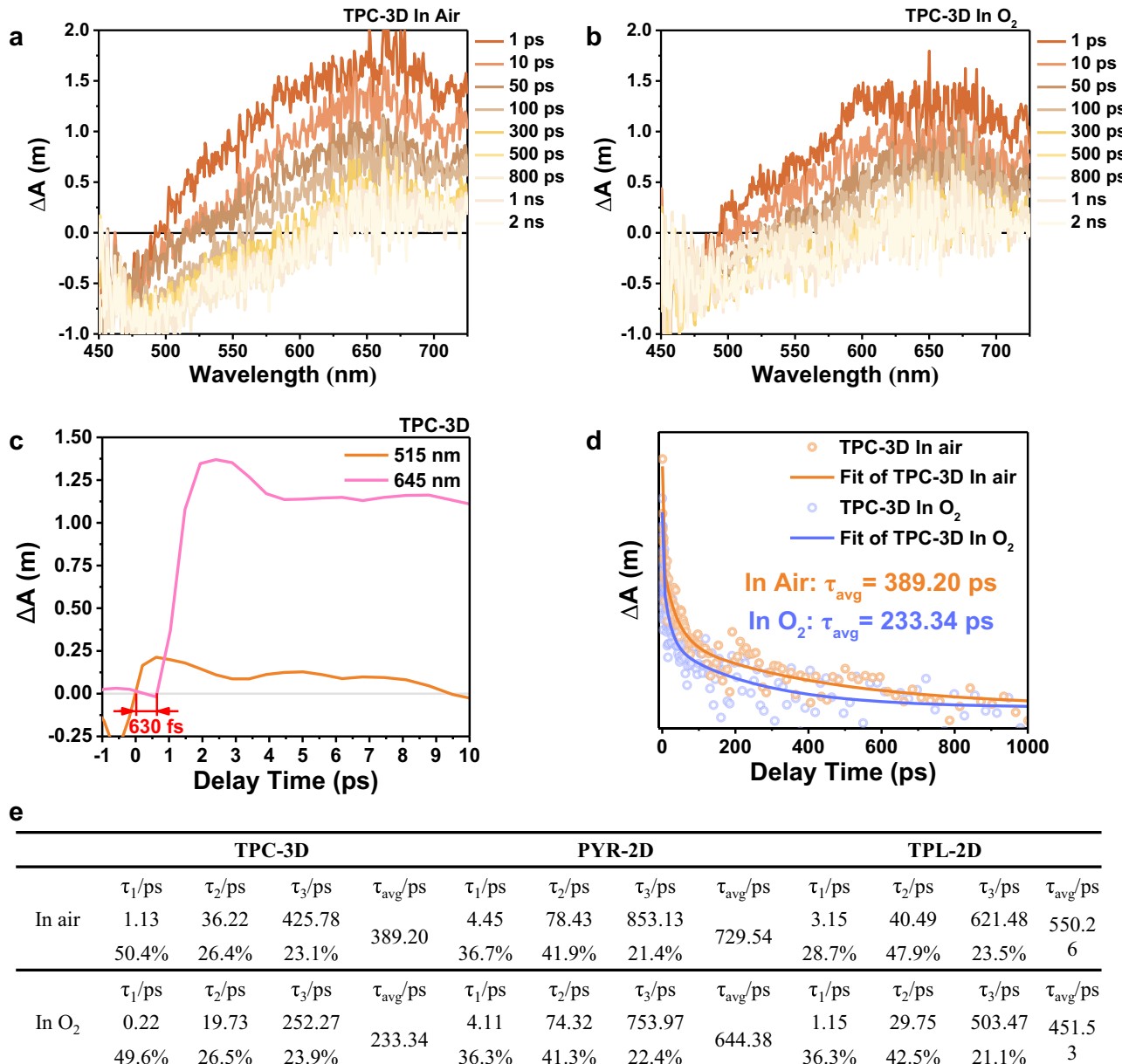

**Fig. 9 | Femtosecond time-resolved transient absorption (TA) measurements of TPC-3D in the visible region. a–b**, Femtosecond TA spectra in 450–780 nm range of TPC-3D under (**a**) air and (**b**) $O_2$ atmosphere. **c** TA kinetic profiles of TPC-3D at 515 nm and 645 nm. TPC-3D exhibited a positive peak consistent with its electron donor monomers (TPC) at ~515 nm, defined as the signal of excitons (Supplementary Fig. 47a). For TPC-3D, the excitons peak at ~515 nm was followed by a positive peak at 645 nm (Supplementary Fig. 47b), attributed to polarons, and the time delay between them was the time for excitons to transfer to polarons, i.e. 630 fs. **d** Comparsion of TA kinetic profiles at 645 nm under air and $O_2$ condition, respectively. **e** Decay lifetimes under $O_2$ and air conditions fitted by the tri-exponential function from corresponding TA kinetic traces at 645 nm.

polaron dissociation in a revers manner, due to the enhance utilization of electrons (Fig. 8). This phenomenon was not observer in 2D structures.

## From excitons dissociation to polarons generation

Building upon the accelerated electron utilization and polaron dissociation discussed above, it was worth exploring whether this precursor effect also contributed to an expedited dissociation of excitons. Additionally, it was important to consider other factors that may influence the rate of excitons dissociation.

As investigated above, both TPC-3D and TPL-2D exhibited significant enhancement in polarons generation rate (630 fs for TPC-3D, 1820 fs for PYR-2D, and 755 fs for TPL-2D) under investigation compared to other reported photocatalysts (Fig. 9c and Supplementary Figs. 48c, 49c)[56]. To elucidate the underlying reasons for this observation, we

initially investigated whether the rapid exciton dissociation was due to an acceleration of polarons separation. By comparing the dissociation rates of excitons in air and oxygen, we verified that the dissociation rates of excitons did not increase in the presence of oxygen, thereby eliminating the possibility of accelerated polarons separation contributing to the acceleration of excitons dissociation (Supplementary Figs. 53, 54).

In order to substantiate the efficient dissociation of excitons, steady-state photoluminescence (PL) emission spectra were recorded, and the photoluminescence quenching yields (PLQYs) were determined to evaluate the overall excitonic recombination efficiencies[57]. The photoluminescence of TPC-3D seemed weaker than that of PYR-2D and TPL-2D (Supplementary Fig. 55). The PLQYs of PYR-2D was 0.11 %, and the PLQYs of TPC-3D and TPL-2D were even below the detectable limit (Supplementary Fig. 56), which indicated that the

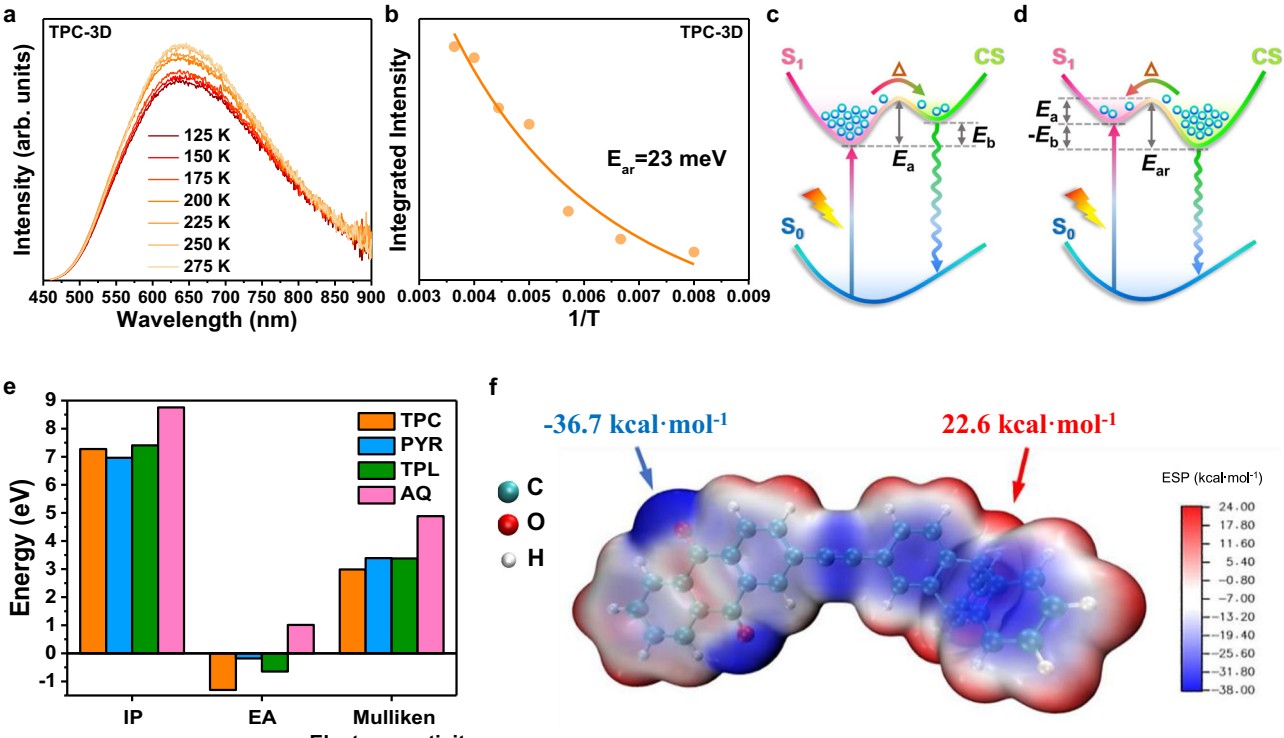

**Fig. 10 | The mechanism of excitonic dissociation. a** Temperature-dependent photoluminescence (TD-PL) of TPC-3D at different temperatures. **b** The $E_{ar}$ was calculated by fitting the temperature dependence of PL intensity with the Arrhenius equation. **c** Illustration of the mutual transitions between the charge separated state (CS) and the lowest singlet excited state ($S_1$) in PYR-2D, where the value of excitonic binding energy ($E_b$) is positive and $E_a$ is the activation energy from $S_1$ to CS. **d** Illustration of mutual transitions between CS and $S_1$ in TPC-3D and TPL-2D, in which the value of $E_b$ is negative, and $E_{ar}$ is the activation energy from CS to $S_1$. This indicated that the energy barrier for exciton dissociation into free charge was lower than the thermal energy at room temperature, and the energy level of the charge-separated state was even lower than that of the exciton state. In other words, the

separation of excitons was spontaneous in TPD-3D and TPL-2D at room temperature. **e** Calculation of the ionization potential (IP), electron affinity (EA), and Mulliken electronegativity of TPC, PYR, TPL and AQ monomers. Mulliken electronegativity = (IP + EA)/2. The smaller the Mulliken electronegativity, the stronger the electron-giving ability. TPC monomer is more capable of giving electrons than PYR and TPL monomers, while AQ monomer tends to gain electrons. **f** Electrostatic potential (ESP) of TPC-3D. The uneven charge distribution of the system is a reflection of the molecular polarity, and the more uneven the distribution results in more positive or negative areas of the electrostatic potential on the surface of the molecule (See Supplementary Fig. 60 for more detailed information).

radiative recombination of excitons was almost completely suppressed in the CPs. In other words, the CPs exhibits ultra-high dissociation efficiency.

In order to further evaluate the mechanism behind the lower PLQYs of TPC-3D and TPL-2D, the activation energy of exciton dissociation ($E_a$) or activation energy of charge recombination ($E_{ar}$) was determined through the temperature-dependent photoluminescence (TD-PL) spectra (Fig. 10a–d and Supplementary Fig. 57)[58,59]. Notably, the TD-PL intensity increased with elevating temperature for TPC-3D and TPL-2D, indicating that $E_a$ was smaller than $E_{ar}$. Thus, the excitons in TPC-3D and TPL-2D could be spontaneously dissociated. The $E_{ar}$ for TPC-3D and TPL-2D could be estimated to be 23 meV and 47 meV, respectively. In contrast, the $E_a$ for PYR-2D was 120 meV. Moreover, the much more production of $^1O_2$ in PYR-2D also confirmed that TPC-3D and TPL-2D exhibited higher excitonic dissociation efficiency (Supplementary Fig. 58a, b). This is because when the excitons could not dissociate efficiently, they would go through spin-flip and transform to triplet excitons, which would react with $O_2$ to generate $^1O_2$ through energy transfer (Supplementary Fig. 58c)[30]. On the other hand, when the excitons dissociated into electrons, $O_2$ tended to generate $\cdot O_2^-$ through electron transfer (Supplementary Fig. 58d).

To further investigate the embedded mechanism behind the spontaneously exciton dissociation, exciton binding energy ($E_b$)

calculations calculation was adopted. $E_b$ has been regarded as a crucial parameter for mediating charge separation in polymeric photocatalysts[60]. The adequate dissociation of exciton in TPC-3D was caused by the remarkable low excitonic binding energies ($E_b$), which indicated that upon photoexcitation, excitons were spontaneously separated into free hole and electron charge carriers after overcoming the Coulomb force (Supplementary Fig. 59)[61]. Moreover, TD-DFT calculation confirmed that exciton dissociation in TPC-3D was more adequate than PYR-2D and TPL-2D, resulting in the sufficient charge aggregation (Supplementary Table 5)[62,63]. In general, *D* index is used to measure the distance between the hole center and electron center, and *t* index is used to measure the separation degree between the hole and the electron from the perspective of charge-separation. Both *D* index and *t* index showed that TPC-3D possessed wider distribution and higher separation degree than that of the excited PYR-2D and TPL-2D, which indicated the better charge-separation ability.

In order to find out the reasons for the lower $E_{ar}$ and $E_b$ of TPC-3D, the electron-giving ability of the three electron-giving unit monomers in the three materials was compared (Fig. 10e), which found that the electron donor of TPC-3D was significantly higher than that of the other two materials, leading to a higher degree of delocalization in TPC-3D. In summary, with the same monomers of the electron acceptor, the stronger electron-giving ability of the electron donor is beneficial to promote the delocalization of the material and the effective separation of excitons, which can be proved by the

electrostatic potential the dipole moment (Fig. 10f and Supplementary Figs. 60, 61).

In conclusion, the sufficient excitonic dissociation in excited TPC-3D was attributed to the stronger delocalization in material itself causing by the stronger electron-giving capacity of TPC moieties as electron donors. All the evidence above demonstrates that the stronger delocalization of TPC-3D itself is the fundamental reason for the expeditious generation of polarons but not the acceleration of polarons separation (Fig. 8).

## Discussion

Our study has established a highly efficient open system that converts solar energy into in-demand chemicals, while elucidating the regulatory mechanisms on accelerated photophysical process through photochemical process. Specifically, we present a self-supporting 3D amorphous organic photocatalyst that achieves a breakthrough efficiency in the photosynthesis of $H_2O_2$ in real water, open air, and at room temperature. The photocatalyst exhibits a remarkable production rate of $H_2O_2$ as high as 9257–9991 $\mu$mol·g$^{-1}$·h$^{-1}$, and a SCC efficiency of 3.6%, surpassing the highest efficiency of typical plants (1%) and representing the exceptional efficiency in artificial photosynthetic systems in the open air.

This photocatalyst's superior performance can be attributed to the effective utilization of internal excitons through intramolecular transfer instead of interlayer transfer. We achieved this by storing electrons within the photocatalyst and exposing the electron storage sites to $O_2$, which facilitates $O_2$ diffusion into the catalyst to extract electrons. As a result, 82.2% of electrons can be utilized through intramolecular electron transfer, which is an order of magnitude faster than interlayer transfer. This approach effectively circumvents electron recombination during interlayer transfer, thereby enabling faster and more efficient utilization of the electrons.

Moreover, of paramount importance is our ability to attain high-performance through deliberate design, coupled with a comprehensive exploration of the kinetics of each step and the corresponding regulatory mechanisms in the photophysical processes. Under ambient conditions, we observed the process from exciton dissociation to polarons generation within 630 fs. Then the polarons rapidly converted to free electrons within 389 ps. The free electrons were ultimately utilized within another 192 ps. Furthermore, the rapid utilization of electrons accelerated the dissociation of polarons without expediting the dissociation of excitons. The expeditious dissociation of excitons in TPC-3D was attributed to the spontaneous dissociation of excitons owning to the strong delocalization of the material (Fig. 8).

Overall, our findings provide important insights into the mechanisms underlying the superior performance of intramolecular charge transfer-based photocatalysts. This study lays a foundation for future construction of higher-performance solar-to-chemical conversion systems and the development of efficient and sustainable energy conversion technologies that utilize solar energy.

## Methods

### Photocatalytic experiments

Photocatalytic $H_2O_2$ production: The suspension contained 1 mg catalysts and 50 mL $H_2O$ was well dispersed by ultrasonication for 30 min. Afterward, the catalyst was irradiated by using a Xe lamp (CEL-HXF300) as the light source. A cutoff filter was used to achieve visible-light irradiation ($\lambda > 400$ nm, average intensity: 100 mW·cm$^{-2}$). The experiment was conducted at room temperature. The sample was filtrated with a 0.22 $\mu$m filter to further remove the photocatalysts.

Determination of $H_2O_2$ concentration: the TMB-$H_2O_2$-HRP enzymatic assay was used to quantify the concentration of $H_2O_2$, where the reaction between $H_2O_2$ and TMB could be instantaneously catalyzed by HRP:

$$H_2O_2 + TMB \xrightarrow{HRP} H_2O + oxTMB \tag{1}$$

The 3,3',5,5'-tetramethylbenzidine (TMB) solution was prepared as follows: 15 mg TMB solid was dissolved in 0.3 mL dimethyl sulfoxide (DMSO), subsequently, 5 mL glycerol solution and 45 mL deionized water containing 20 mg ethylene diamine tetraacetic acid (EDTA) and 95 mg citric acid was added.

The horseradish peroxidase (HRP) solution was prepared as follows: dissolve 2 mg peroxidase (from horseradish) in 10 mL deionized water.

To obtain the calibration curve, a known concentration of $H_2O_2$ (as the hydrogen donor) was added to the TMB solution. It should be noted that the concentration of HRP was maintained at 12.5 $\mu$g·mL$^{-1}$. After 3 min, added 10 $\mu$L concentrated hydrochloric acid and measured by a UV-visible spectrophotometer at 450 nm to obtain a standard calibration curve. Then the $H_2O_2$ concentration of the samples could be quantified according to the linear relationship between signal intensity and $H_2O_2$ concentration.

### SCC efficiency measurements

The solar-to-chemical energy conversion (SCC) efficiency was measured by photocatalytic experiments employing an AM 1.5 G solar simulator as the light source (100 mW·cm$^{-2}$). The concentration of catalyst is 0.4 g·L$^{-1}$. Calculated the SCC efficiency ($\eta$) via the following equation:

$$\eta(\%) = \frac{\Delta G(H_2O_2) \times n(H_2O_2)}{t_{ir} \times S_{ir} \times I_{AM}} \times 100\% \tag{2}$$

In the equation, $\Delta G(H_2O_2) = 117$ kJ·mol$^{-1}$, is the free energy for $H_2O_2$ generation. $n(H_2O_2)$ is the amount of $H_2O_2$ generated during the photocatalytic reaction. The irradiation time $t_{ir}$ is 3600 s, and the irradiated sample area ($S_{ir}$) is $1 \times 10^{-4}$ m$^2$. $I_{AM}$, the total irradiation intensity of the AM 1.5 global spectra (300 nm–2500 nm), is 100 mW·cm$^{-2}$.

### Transient absorption spectroscopy measurements

TA spectra of TPC-3D, PYR-2D, and TPL-2D w ere measured on a Helios femtosecond transient absorption spectrometer (Helios Fire, Ultrafast Systems, LLC). A 400 nm pump pulse was generated via an optical parametric amplifier (OPerA Solo, Coherent). The 400 nm laser intensity was 50 $\mu$W. The preparation of samples was as follows: 10 mg of photocatalyst was dispersed in 50 mL deionized water. Then the suspension was subjected to ultrasonication by an ultrasonic cell pulverizer (SCIENTZ-II D) for 48 h. Afterward, the suspension was left overnight to allow the large particles to settle down while the fine particles were stably suspended in water.

Measurements were performed using a cuvette with stopper. For TA spectra in air atmosphere, the suspension was added directly to the cuvette and tested under open conditions. For TA spectra in oxygen atmosphere, oxygen was injected into the suspension and bubbled for 20 min to saturate it, and then the test was performed with the stopper closed. When measuring TA spectra with the addition of sacrificial agents, the concentration of sacrificial agents was maintained at 10 mM.

### Computational methods

The Density functional theory (DFT) calculation was conducted as implemented in the Gaussian 09 D.01 program package[64] and Grimme-D3 dispersion correction was used[65]. GaussView6 was employed for visualization. The geometry optimization and frequency analysis were performed at the B3LYP/6-31 G(g,d) level of theory[66]. The atomic coordinates of the optimized computational models were shown in

Supplementary Table 6–8. The optimized structures were utilized to calculate single-point energy with PBE0/ma-TZVP[66–68]. The convergence criteria for energy, maximum force, root mean square force, maximum displacement, and root mean square displacement was $10^{-6}$ Hartree, $4.5 \times 10^{-4}$ Hartrees/Bohr·Radians, $3 \times 10^{-4}$ Hartrees/Bohr·Radians, $1.8 \times 10^{-3}$ Bohr·Radians, $1.2 \times 10^{-3}$ Bohr·Radians respectively. Time-dependent density functional theory (TD-DFT) was carried out at the PBE0/6-31 G(d,p) level of theory and applied to the investigation of the transfer direction of electrons[69–71]. Analysis and visualization of hole and electron distribution were performed by Multiwfn[62]. Exciton binding energy was calculated by the def2-SVP basis set[58]. The electrostatic potential and Mulliken electronegativity involved in the analysis were evaluated by Multiwfn based on an efficient algorithm and filled with colors using VMD1.9.3[58,62,63,72].

The adsorption energy ($E_{adsorb}$) of $O_2$ molecule on the surface is calculated as follow:

$$E_{adsorb} = E_{total} - E_{surface} - E_{O_2} \qquad (3)$$

where $E_{total}$ represents the energy of surface with adsorbed $O_2$ molecule, $E_{surface}$ and $E_{O_2}$ represent the energies of isolated surface and $O_2$, respectively.

## Data availability

The data that support the findings of this study are available from the corresponding author upon request.

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

## Acknowledgements

This work was supported by the National Natural Science Foundation of China 22206209 (Y.-X.Ye), 22336007 (G.F.O.Y.), the Natural Science Foundation of Guangdong Province 2022A1515011953 (Y.-X.Ye), the Guangdong Basic Research Center of Excellence for Functional

Molecular Engineering Project 31000-42080002 (G.F.O.Y.), and Southern Marine Science and Engineering Guangdong Laboratory (Zhuhai) via project No. SML2023SP220 (Y.-X.Ye). The authors thank the PL group of Instrumental Analysis & Research Center, Sun Yat-sen University for their contributions to temperature-dependent PL and PLQYs measurements.

## Author contributions

Y.Y.H., Y.-X.Ye and G.F.O.Y. co-proposed the idea and designed the experiments. Y.Y.H. contributed to perform the experiments. Y.Y.H. and M.H.S. performed the DFT calculations. Y.Y.H., Y.-X. Ye, M.H.S., H.J.Y., Y.G.H., J.Q.X., F.Z. and X.Y. contributed to performing and analyzing the experiments. Y.Y.H., Y.-X.Ye and G.F.O.Y. participated in writing the manuscript.

## Competing interests

The authors declare no competing interests.
