## [Peer Review File · Nature Communications]

Achieving a solar-to-chemical efficiency of 3.6% in ambient conditions by inhibiting interlayer charges transportREVIEWER COMMENTS

Reviewer #1 (Remarks to the Author):

The manuscript by Huang and co-workers reports on semiconductor polymer photocatalysts for hydrogen peroxide photoproductions. The author report the synthesis of three polymeric photocatalysts, which mainly differ in the 2D or 3D structure, their characterisation and photocatalytic response. Notably, one of the polymers was reported to achieve high apparent quantum yields and solar-to-chemical efficiency. In my opinion, the manuscript could be of interested to the readers of Nature Communications, but after major revision.

- My main concern is related to the polymer stability and I was not convinced by the evidence presented that the “component” was maintained for 5 cycles. Supplementary Figure 16 clearly shows the appearance of a new peak around 1700 cm^{-1} even after 1 cycles and clearly increases even more after 5 cycles. In a similar fashion, the alkynyl peaks decreases intensity. Together, this could be takes as evidence of photooxidation of polymers. After all, it has been reported that semiconductor photocatalysts for H_2O_2 production undergo self-oxidation (J. Am. Chem. Soc. 2019, 141, 22, 9063–9071). The authors further stress the interaction between the alkynyl group and O_2 , so it is likely some oxidation occurs there.

- Is there a particular reason the post-catalytic characterisation has been performed after 5 cycles, even though the materials was used for at least 15 cycles?

- While charge accumulation on anthraquinone is intriguing, the only experiment from which this in inferred is reported in Figure 3c. However, in this Figure, as well as in the rest of the text, apparently no duplicates (or triplicates) are reported and is unclear what the error of these measurements are. Without a duplicate (at least) the claim of dark production of H_2O_2 could be ascribed to an error. I would also expect the material to change color upon charge accumulation and perhaps this can be tested also in (spectro)electrochemistry.

- Anthraquinones/hydroquinones are very well know for their use in production of H_2O_2 from O_2 . The mechanism proposed by the authors does not utilise this property of anthraquinone, but proposes that hydroquinone gives one electron to form superoxide (which is quite unexpected), and I believe a discussion is warranted.

- For water oxidation claims, perhaps it should be tested through isotope labelling.

- I would also suggest that graphs such as Fig 2d and Fig 3c have reported in the second y-axis the amount of H_2O_2 as concentration.

- Another point that warrants discussion is that polymer photocatalysts have been reported to interact directly with O_2 , by transferring electrons to O_2 to form superoxide. The authors should perhaps discuss why their systems is different and how has the interactions of donor excited state with O_2 to form superoxide has been excluded.

- Another aspect of not is the performance of the photocatalyst in “real water”. In my opinion, this should be better explained. A counter argument could be that it is expected that the H_2O_2 produced in presence of organics should be lower, since superoxide has been already reported in organic pollutant photo degradation.

- I would also suggest a change to the title and abstract. Especially the title (but also the abstract) is very focused on performance so it is not immediately clear what has actually been done.

- Some parts of the manuscript are over-hyped (see also comment above). For example it is claimed that this is the first instance where SCC has surpassed plants, but just this year has been reported a 7% water splitting efficiency (Nature (<https://www.nature.com/>) 613, pages 66–70 (2023)). Perhaps the authors meant the first instance of a polymeric semiconductor for solar to H_2O_2 production?

- Also in the introduction the authors mention that a series of newly develop in situ transient characterisation techniques has been develop. However, in the later discussion, it is not clear what has been actually develop and what the new techniques are.
- Since the system reported here by the authors is closely related to other polymer semiconductors for H₂O₂ production, perhaps a discussion on polymer design and the importance of each part is important, also to have a proper comparison to closely-related similar systems.

Reviewer #2 (Remarks to the Author):

This manuscript reports a polymeric photocatalyst achieving a solar-to-H₂O₂ conversion efficiency of 3.6% under ambient conditions. The performance can be attributed to the rapid intramolecular electron transfer and the spontaneous dissociation of excitons. However, the manuscript should be further improved in terms of the following issues before publication:

Comments and Suggestions for Authors:

Q1. On page 9, the authors declared that “DFT calculation revealed that O₂ was dominantly chemisorbed on the alkynyl moieties to form the endoperoxide species in TPC-3D, rather than the AQ or triptycene (TPC) moieties.” Is there any direct evidence that oxygen is chemically adsorbed in the alkynyl group of TPC-3D? Does the alkynyl group’s Raman or FTIR peak change during the photocatalytic reaction as oxygen is stored? Can it be observed by in situ experiments? For example, in situ Raman or in situ FTIR.

Q2. On pages 15 and 16, the authors claimed that E_a is the activation energy of charge recombination and the reverse potential barrier, respectively. I wonder if these two identical abbreviations mean the same thing.

Q3. The authors tested the EPR spectra of photocatalysts under ambient conditions in light and darkness, thus claimed that the results show that AQ can store electrons. Because electrons stored in the structure can undergo recombination or ORR involving oxygen and water in the dark, the EPR tests described in the paper does not appear to represent the electronic signal during the real reaction process. It is recommended to test the EPR spectra of the material under reaction conditions. In addition, what happened to the EPR signal once the trapped electrons finished the reaction in the dark?

Q4. Please check the legend carefully for mistakes. For example, TCL-2D was incorrectly spelled as TCL-3D in supplementary 19a.

Q5. The authors claimed that the electrons stored under dark conditions come from electron-hole pairs produced by light excitation. Since electrons and holes are formed in pairs, what process do the holes in TPC-3D go through during the reaction (from light to darkness)?

Q6. The specific surface area is analyzed through the BET, however, the catalytic active area is not equal to the specific surface area. Thus, the analysis of real catalytic active area should be added.

Q7. The real water, such as lake water, river water, and seawater, is used for photocatalytic reaction. This is very interesting, but, the element contents of these real water is not mentioned. And how the extra elements in real water affect the photocatalytic performance?

Q8. During the catalytic reaction, the analysis of adsorption-conversion process seems missing, and this should be discussed by DFT or in situ technique.

Q9. During the catalytic reaction, did the amorphous carbon make contribution to the performance? How?

Q10. In Fig. S7 and Fig. S16, complete XRD data of 3-10 degrees should be provided.

Q11. There are some support information graphs that are not mentioned in the article, such as S20 and S43. Please check carefully.

Q12. How is the stability of TPC-3D in pure water and real water (including lake water, river water, and sea water)? Is it only limited to 5 hours or 5 cycles?

Response to Reviewers

We are deeply appreciative of the editor and reviewers whose rigorous assessment has substantially elevated our manuscript's academic rigor and lucidity. The review process was thorough and the feedback provided both insightful and constructive, driving significant enhancements in our work's presentation and substance. Notably, we have undertaken a thorough examination of stability-related concerns, delineating potential causes and implementing solutions. Every comment has been scrupulously considered; revisions are distinctly annotated in yellow within the amended manuscript. Moreover, we have furnished a detailed, point-by-point riposte to the reviewer's queries, distinguished in blue, for unambiguous identification in the appended document.

Reviewer #1

The manuscript by Huang and co-workers reports on semiconductor polymer photocatalysts for hydrogen peroxide photoproductions. The author report the synthesis of three polymeric photocatalysts, which mainly differ in the 2D or 3D structure, their characterisation and photocatalytic response. Notably, one of the polymers was reported to achieve high apparent quantum yields and solar-to-chemical efficiency. In my opinion, the manuscript could be of interested to the readers of Nature Communications, but after major revision.

Response:

We sincerely appreciate the reviewer's constructive comments on our manuscript. These comments are invaluable and have greatly assisted us in revising and enhancing our paper, while also providing important guidance for our ongoing research. We have carefully reviewed each comment and have made revisions accordingly, with the hope that they will meet with your approval.

Q1. My main concern is related to the polymer stability and I was not convinced by the evidence presented that the “component” was maintained for 5 cycles. Supplementary Figure 16 clearly shows the appearance of a new peak around 1700 cm^{-1} even after 1 cycles and clearly increases even more after 5 cycles. In a similar fashion, the alkynyl peaks decreases intensity. Together, this could be takes as evidence of photooxidation of polymers. After all, it has been reported that semiconductor photocatalysts for H_2O_2 production undergo self-oxidation (J. Am. Chem. Soc. 2019, 141, 22, 9063–9071). The authors further stress the interaction between the alkynyl group and O_2 , so it is likely some oxidation occurs there.

Response:

We acknowledge the existence of partial oxidation in the photocatalytic process and have pinpointed the specific site undergoing oxidation during the photocatalytic cycling. This specific site has a minor contribution to the overall production of H_2O_2 in the photocatalytic process. Consequently, oxidation at this site has a modest impact

on the photocatalytic efficiency. Furthermore, we also proposed a method to decelerate the onset of the oxidation phenomenon.

Fig. R1. **a**, Recycle experiments of CPs in pure water. **b**, Chemical components of TPC-AQ. **c**, The FT-IR spectra before and after 5 cycles of TPC-AQ. **d–e**, EPR spectra of **(d)** TPC-3D and **(e)** TPC-AQ by using DMPO as a scavenger in O₂ atmosphere under visible light irradiation to examine the existence of ·O₂⁻ during ORR.

Firstly, we identified the site undergoing oxidation and the substances that caused oxidation. The FT-IR spectroscopy of the material after cycling revealed a decrease in the alkynyl peak, along with the appearance of a new peak near 1700 cm⁻¹. This indicated that self-oxidation of the alkynyl sites might occur during the photolysis process. There was indeed a slightly downward trend in the yield of H₂O₂ with an increasing number of cycles (Fig. R1a). To further substantiate the occurrence of oxidation at alkynyl, we synthesized a polymer devoid of alkynyl groups using triptycenes as the electron donor and anthraquinone as the electron acceptor, designated as TPC-AQ (Fig. R1b). Remarkably, the infrared spectrum of TPC-AQ without alkynyl group exhibited no significant changes after five cycles (Fig. R1c). It is inferred that some strong oxidizing substances were generated on the alkynyl during the photocatalytic process, which led to the self-oxidation of alkynyl.

In order to identify the species causing oxidation, the electron paramagnetic resonance (EPR) spectra measurement was conducted. As depicted in Fig. R1d–e, TPC-3D containing alkynyl group exhibit the typical six-line characteristic peaks of DMPO·O₂⁻ using 5, 5-dimethyl-1-pyrroline N-oxide (DMPO) as the spin-trap agent (*Angew. Chem. Int. Ed.* 2023, **62**, e202305355). In contrast, TPC-AQ without alkynyl group had no corresponding peak. The above phenomenon indicates that the production of H₂O₂ at alkynyl sites involves a two-step 2e⁻ oxygen reduction reaction

(ORR) process with $\cdot\text{O}_2^-$ as the intermediates. The strong oxidizing nature of $\cdot\text{O}_2^-$ likely leads to its attack on the alkynyl, causing the observed changes in the infrared spectrum.

Nonetheless, this partial oxidation exerts a minor effect on the production of H_2O_2 , resulting in only a slight reduction in its yield (Fig. R1a). This is attributed to the relatively small percentage of ORR occurring at the alkynyl group. To investigate the percentage of ORR occurring on alkynyl, superoxide dismutase (SOD) was used as a scavenger of $\cdot\text{O}_2^-$ to inhibit two-step $2e^-$ ORR process on alkynyl (Fig. R2a). In the presence of SOD, the H_2O_2 yield of TPC-AQ remained largely unchanged, whereas that of TPC-3D decreased by less than one-fifth which represented the proportion of ORR on the alkynyl. Thus, TPC-3D maintains high performance after five cycles.

To solve the problem of self-oxidation, we immobilized TPC-3D on a glass slide to construct a device for the cycle experiment (Fig. R2b). TPC-3D exhibited a stable H_2O_2 yield over 15 cycles (Fig. R2c). The FT-IR Spectroscopy displayed that its alkyne peaks were still clearly visible after 15 cycles (Fig. R2d), and there were no significant changes in other characterizations (Fig. R2e–g).

Relevant instructions have been added in revised manuscript at page 7 and Supplementary Note 2, and additional data have been added to the Supplementary Fig. 17.

Fig. R2. **a**, H_2O_2 yield of TPC-3D and TPC-AQ before and after adding SOD. **b**, Immobilization of TPC-3D powders (2 mg) onto a glass slide (20 mm × 20 mm) for H_2O_2 production in 400 mL of lake water under xenon-lamp irradiation ($\lambda > 400$ nm, light intensity: $100 \text{ mW}\cdot\text{cm}^{-2}$). **c**, The stable H_2O_2 production activity of TPC-3D under the condition described in Fig. R2b. **d–e**, Comparison before and after 15 cycles of TPC-AQ under the condition described in Fig. 2b: **(d)** FT-IR spectra, **(e)** PXRD image. **f–g**, The **(f)** SEM image and **(g)** TEM image after 5 cycles under the condition described in Fig. 2b.

Q2. Is there a particular reason the post-catalytic characterisation has been performed

after 5 cycles, even though the materials was used for at least 15 cycles?

Response:

Fig. R3. a, Recycle experiments of CPs in pure water. b–c, Comparison between the as-prepared TPC-3D and TPC-3D after 5 cycles of photoreaction: (a) XRD image, (b) FT-IR spectra. c–d, The (c) TEM image and (d) SEM image after 5 cycles of recycling.

The 5 cycles were carried out with the catalyst in a dispersed state within pure water (Fig. R3), whereas the 15 cycles involved the catalyst being immobilized on glass slide using Nafion (Fig. R4a, b). In the original manuscript, we recovered the catalyst that had undergone five cycles while dispersed in water and proceeded to characterize it (Fig. R3b–e). This method was limited to 5 cycles due to the considerable catalyst loss during recovery and reuse, which could not guarantee a sufficient quantity of catalyst for additional cycles. For the fifteen cycles with the catalyst immobilized on glass slides, characterization was not feasible without damaging the device. Moreover, the presence of Nafion could potentially alter the characterization outcomes. For these reasons, data following the 15 cycles have not been provided in the original manuscript.

In consideration of the reviewers' concerns regarding stability, we removed the materials from the glass slide after 15 cycles and found a solvent capable of eluting Nafion. Following this treatment, characterization was performed. The FT-IR Spectroscopy displayed that its alkyne peaks were still clearly visible after 15 cycles (Fig. R4c), with no significant changes observed in other characterizations (Fig. R4d–f). Relevant instructions had been added in revised manuscript at page 7 and Supplementary Note 2, and additional data have been added to the Supplementary Fig. 17.

Fig. R4. **a**, Immobilization of TPC-3D powders (2 mg) onto a glass slide (20 mm × 20 mm) for H₂O₂ production in 400 mL of lake water under xenon-lamp irradiation ($\lambda > 400$ nm, light intensity: 100 mW·cm⁻²). **b**, The stable H₂O₂ production activity of TPC-3D under the condition described in Fig. R4a. **c–d**, Comparison before and after 15 cycles of TPC-AQ under the condition described in Fig. 2a: (**c**) FT-IR spectra, (**d**) PXRD image. **e–f**, The (**e**) SEM image and (**f**) TEM image after 5 cycles under the condition described in Fig. R2a.

Q3. While charge accumulation on anthraquinone is intriguing, the only experiment from which this is inferred is reported in Figure 3c. However, in this Figure, as well as in the rest of the text, apparently no duplicates (or triplicates) are reported and it is unclear what the error of these measurements are. Without a duplicate (at least) the claim of dark production of H₂O₂ could be ascribed to an error. I would also expect the material to change color upon charge accumulation and perhaps this can be tested also in (spectro)electrochemistry.

Response:

In response to your invaluable feedback, we replicated the experiment of dark production of H₂O₂ for two additional times, incorporating error bars in the resulting data (Fig. R5a). The repeated experiments confirmed that the generation of H₂O₂ persisted for a duration of about 20 min in dark. These phenomena indicated that the electrons were stored in AQ, which then reduce O₂ to generate H₂O₂ for about 20 min.

In addition, significant color change during the photocatalytic process can be regarded as another visual indicator of the accumulation of electrons. Upon irradiation in air atmosphere, the material changed from a brown to an orange-yellow, whereas in an argon atmosphere, the material further shifts to a bright yellow, which is attributed to the accumulation of electrons (Fig. R5b). And in the UV/Vis spectra (Fig. R5b), the increase of absorbance indicated the formation of the reduced species (*Angew. Chem. Int. Ed.* 2020, 59, 10904–10908), which was confirmed by the in situ diffuse reflectance infrared Fourier transform spectroscopy (DRIFTS). In DRIFTS as shown

in Fig. R5c, the peaks at 1358, 1482 and 1680 cm^{-1} were assigned to the one-electron accumulation state of AQ (AQH), two-electron accumulation state of AQ (AQH₂) and AQ, respectively (*Proc. Natl. Acad. Sci. U. S. A.* 2021, **118**, e2115666118). With the progression of irradiation, the peak of AQ diminished in argon, along with the emergence of peaks corresponding to the reduced species AQH and AQH₂. Subsequently, upon the removal of the light source and the injection of oxygen, the peaks of AQH and AQH₂ decreased while the peak of AQ increased, indicative of the reaction between the electron-storage reduced species and O₂. These phenomena provide strong evidence for the electron storage capability of anthraquinone.

The discussion of electrons accumulation on anthraquinone was added in revised manuscript at pages 11–12.

Fig. R5. **a**, H₂O₂ yields through ORR at AQ moieties. Photosynthesis was conducted in an argon atmosphere for one hour, followed by immediate injection of pure O₂ into the photocatalytic system after stopping the illumination. **b**, Absorption spectra of TPC-3D samples before and after illumination in argon and air. **c**, In-situ DRIFTS of light in argon and dark in oxygen process of TPC-3D.

Q4. Anthraquinones/hydroquinones are very well known for their use in production of H₂O₂ from O₂. The mechanism proposed by the authors does not utilise this property of anthraquinone, but proposes that hydroquinone gives one electron to form superoxide (which is quite unexpected), and I believe a discussion is warranted.

Response:

We apologize for the unclear expression that led to a misunderstanding by the reviewer regarding the oxygen reduction reaction (ORR) pathway on AQ. The pathway mentioned by the reviewer involving the formation of the superoxide via single electron transfer actually occurred on the alkynyl, not on AQ. We would like to offer the following clarification:

Fig. R6. **a**, EPR spectra of TPC-3D by using DMPO as a scavenger in O₂ atmosphere under visible light irradiation to examine the existence of $\cdot\text{O}_2^-$ during ORR. **b**, Chemical components of TPC-AQ. **c**, EPR spectra of TPC-AQ by using DMPO as a scavenger in O₂ atmosphere under visible light irradiation to examine the existence of $\cdot\text{O}_2^-$ during ORR. **d**, H₂O₂ yield of TPC-3D and TPC-AQ before and after adding SOD.

There are two distinct pathways for ORR: one occurs on the alkynyl where can adsorb O₂, and then $\cdot\text{O}_2^-$ intermediates are formed through a single-electron transfer, subsequently yielding H₂O₂. The other pathway involves the storage of two electrons on AQ, which then reacts with O₂ to produce H₂O₂ without involving the $\cdot\text{O}_2^-$ intermediates.

To elucidate the ORR pathway, electron paramagnetic resonance (EPR) spectra were conducted to detect $\cdot\text{O}_2^-$ with 5, 5-dimethyl-1-pyrroline N-oxide (DMPO) as the spin-trap agent (*Angew. Chem. Int. Ed.* 2023, **62**, e202305355). As depicted in Fig. R6a, TPC-3D containing both alkynyl and AQ exhibited the typical six-line characteristic peaks of DMPO- $\cdot\text{O}_2^-$, indicative of the presence of superoxide. For comparison, we synthesized a polymer devoid of alkynyl groups using triptycenes as the electron donor and anthraquinone as the electron acceptor, designated as TPC-AQ (Fig. R6b). It is noteworthy that TPC-AQ, which contains AQ but not alkynyl, did not exhibit the peak of DMPO- $\cdot\text{O}_2^-$ (Fig. R6c). The comparison suggested that ORR pathway occurred on the alkynyl via a two-step $2e^-$ process involving the formation of the $\cdot\text{O}_2^-$ intermediates, while the ORR pathway on AQ reduced O₂ to produce H₂O₂ via a one-step $2e^-$ process bypassing $\cdot\text{O}_2^-$. To further support this conclusion, we conducted a photocatalysis experiment with the addition of superoxide dismutase (SOD), which acts as a scavenger of $\cdot\text{O}_2^-$. The presence of SOD did not significantly affect the H₂O₂ yield of TPC-AQ, while it reduced the yield in TPC-3D by about 19% (Fig. R6d). This differential response to SOD treatment confirms our hypothesis regarding the distinct ORR pathways on the alkynyl and AQ moieties.

Fig. R7. **a**, Schematic representation of the anthraquinone (AQ) oxidation process. Reprinted from *ACS Catal.* 2018, **8**, 4064–4081. **b**, Key steps of H_2O_2 production by two-electron ORR process on AQ. **c**, In-situ DRIFTS of light in argon and dark in oxygen process of TPC-3D.

Additionally, the ORR on AQ of TPC-3D is fundamentally similar to the traditional applications in which anthraquinones/hydroquinones react with O_2 to produce H_2O_2 . The difference is as below: the conventional industrial AQ oxidation method uses palladium (Pd) as a catalyst to hydrogenate AQ in the presence of hydrogen gas, forming hydroquinone (AQH_2), which then generates H_2O_2 (Fig. R7a) (*ACS Catal.* 2018, **8**, 4064–4081). In contrast, our system utilizes only sunlight to activate the photocatalyst, generating photo-induced charges that drive the formation of AQH_2 , and subsequently reacts with oxygen to H_2O_2 (Fig. R7b).

Specifically, as shown in Fig. R7b, under visible-light irradiation, the electron donor generated an electron and transferred to AQ, where it reacted with the $\text{C}=\text{O}$ double bond to form C-O oxygen-centered organic radicals, and combined with proton released from the H_2O molecule initiated an electron-coupled hydrogenation of AQ to form AQH. Subsequently, H_2O provided another proton and an electron, resulting in the formation of a two-electron storage of anthrahydroquinone (AQH_2) (*Proc. Natl. Acad. Sci. U. S. A.* 2021, **118**, e2115666118; *ACS Catal.* 2022, **12**, 12954–12963). Then, AQH_2 reacted with O_2 to generate AQH_2 -1,4-endoperoxide, which subsequently coupled the adjacent hydrogen in the hydroxyl group to release H_2O_2 and regenerated the AQ redox center (*Nat. Commun.* 2024, **15**, 2649). The in situ diffuse reflectance infrared Fourier transform spectroscopy (DRIFTS) in Fig. R7c demonstrated the transition from AQ (1680 cm^{-1}) to AQH (1358 cm^{-1}), AQH_2 (1482 cm^{-1}) under irradiation (*Proc. Natl. Acad. Sci. U. S. A.* 2021, **118**, e2115666118). Furthermore, with the injection of O_2 , a significantly enhanced infrared vibration peak

located at 910 cm^{-1} could be observed which was assigned to the 1,4-endoperoxide intermediate species (*Angew. Chem. Int. Ed.* 2023, **62**, e202305355), further corroborating the ORR on AQ.

The discussion about the oxygen reduction reaction process on AQ was added in revised manuscript at pages 10–12.

Q5. For water oxidation claims, perhaps it should be tested through isotope labelling.

Response:

Thank you for your valuable feedback. The isotopic experiment was conducted as follows. Specifically, 5 mg of photocatalyst was ultrasonically dispersed into 1 mL of H_2^{18}O , which was purged with O_2 for 5 minutes. Then, the suspension was illuminated for 2 h by a 300 W Xe lamp ($\lambda > 400\text{ nm}$, $100\text{ mW}\cdot\text{cm}^{-2}$) at ambient temperature. The post-reaction photocatalytic hydrogen peroxide solution was transferred to a new reactor, and Ar was injected to eliminate O_2 . MnO_2 was added to catalyze the decomposition of hydrogen peroxide, and the evolved oxygen gas was analyzed through GC-MS (Agilent 8860-5977B).

From Fig. R8, it was evident that, when isotopically labeled water (H_2^{18}O) was employed in the photocatalytic reaction instead of conventional water (H_2^{16}O), and the subsequently generated hydrogen peroxide was converted to O_2 by MnO_2 , a distinct enhancement of the isotopic signal peak corresponding to $^{18}\text{O}_2$ was observed. This observation effectively demonstrated the evolution of H_2O_2 by hole-induced water oxidation reaction (WOR) pathway. We have provided the discussion in the revised manuscript at page 10 and in Supplementary Fig. 28.

Fig. R8. H_2^{18}O isotope experiment to explore the H_2O_2 evolution through WOR pathway for TPC-3D. Before the addition of MnO_2 , there were weak $^{18}\text{O}_2$ and $^{16}\text{O}_2$ signals in the system were weak, which was attributed to trace of O_2 from the reaction environment.

Q6. I would also suggest that graphs such as Fig 2d and Fig 3c have reported in the second y-axis the amount of H_2O_2 as concentration.

Response:

We thank the reviewer for the suggestion. H_2O_2 concentration had been labeled

on the second y-axis of Fig. 2d and 3c. However, we prefer to use the H_2O_2 yield ($\mu\text{mol}\cdot\text{g}^{-1}\cdot\text{h}^{-1}$) as a benchmark because in actual production processes, the catalyst concentration is often kept low due to cost considerations. In our study, we mimicked practical applications by using a low catalyst amount, in large water volumes, and under ambient conditions. Thus, referencing the H_2O_2 yield ($\mu\text{mol}\cdot\text{g}^{-1}\cdot\text{h}^{-1}$) more accurately reflects the advantages of our system.

Fig. 2. d, The stable H_2O_2 production of TPC-3D under the following condition: immobilization of TPC-3D powders (2 mg) onto a glass slide (20 mm × 20 mm) for H_2O_2 production in 400 mL of lake water under xenon-lamp irradiation ($\lambda > 400$ nm, light intensity: $100 \text{ mW}\cdot\text{cm}^{-2}$)

Fig. 3. c, H_2O_2 yields through ORR at AQ moieties. Photosynthesis was conducted in an argon atmosphere for one hour, followed by immediate injection of pure O_2 into the photocatalytic system after stopping the illumination.

Q7. Another point that warrants discussion is that polymer photocatalysts have been reported to interact directly with O_2 , by transferring electrons to O_2 to form superoxide. The authors should perhaps discuss why their systems is different and how has the interactions of donor excited state with O_2 to form superoxide has been excluded.

Response:

As discussed in question 4, there are two distinct pathways for ORR: one occurs on the alkynyl where it can spontaneously adsorb O_2 (Fig. R9a), and then $\cdot\text{O}_2^-$ intermediates are formed through a single-electron transfer, subsequently yielding

H₂O₂ (Pathway I). The other pathway involves the storage of two electrons on AQ, which then reacts with O₂ to produce H₂O₂ without involving the $\cdot\text{O}_2^-$ intermediates (Pathway II). These two pathways coexist, and which pathway occurs depends on whether oxygen is pre-adsorbed or not. When oxygen is pre-adsorbed on the alkynyl groups, Pathway I will occur; when there is no pre-adsorbed oxygen on the alkynyl groups, electrons will be rapidly stored on AQ through intramolecular transfer to form AQH₂, and subsequently Pathway II will occur.

Additionally, unlike the pathway where electrons are directly transferred to pre-adsorbed O₂ forming superoxide, the ORR pathway on AQ first stores electrons to form AQH₂, which then reacts with O₂. DFT calculations have revealed that this is due to the difficulty of oxygen pre-adsorbing on AQ (Fig. R9b). However, once electrons are stored to form AQH₂, the energy barrier for oxygen adsorption is significantly reduced (Fig. R9c). Subsequently, AQH₂ with two-stored electrons undergoes a one-step two-electron ORR to produce H₂O₂, bypassing the $\cdot\text{O}_2^-$ intermediates.

Fig. R9. a–d, Molecular model and adsorption energy (E_{adsorb}) for TPC-3D when oxygen is adsorbed at (a) alkynyl moieties and (b) AQ moieties. c, Molecular model and adsorption energy (E_{adsorb}) for excited TPC-3D when oxygen is adsorbed at AQH₂ moieties.

Regarding the interaction between the excited donor and O₂: first, before excitation, DFT calculations indicate that pre-adsorption on the donor is difficult due to high adsorption energy (Fig. R10a–c). Upon illumination, and in the absence of pre-adsorbed oxygen on the donor, the excited photocatalysts led to the localization of the photo-induced holes at electron donor, while the electrons at the carbonyl oxygen and carbonyl carbon (Fig. R10d). This means that electrons are rapidly stored on AQ through intramolecular transfer, leaving the holes on the donor. Since the formation of superoxide requires the acquisition of electrons, the electron-deficient excited electron donor at this point is more inclined to obtain electrons from the water oxidation reaction, rather than losing electrons to undergo an oxygen reduction reaction with O₂.

Therefore, the possibility of the excited electron donor forming superoxide with O₂ is relatively small. We have provided the discussion in the revised manuscript at pages 12–13 and in Supplementary Note 5.

Fig. R10. a–c, The adsorption energy when oxygen is adsorbed in different positions of (a) TPC-3D, (b) PYR-2D and (c) TPL-2D. d, Distribution of holes (blue) and electrons (green) in TPC-3D obtained through TD-DFT calculations (Isosurface value = 0.001).

Q8. Another aspect of not is the performance of the photocatalyst in “real water”. In my opinion, this should be better explained. A counter argument could be that it is expected that the H₂O₂ produced in presence of organics should be lower, since superoxide has been already reported in organic pollutant photo degradation.

Response:

Fig. R11. H₂O₂ yield of TPC-3D and TPC-AQ before and after adding SOD.

Organic pollutants can indeed react with superoxide radicals, which typically leads to a decrease in the production of H₂O₂ (*Angew. Chem. Int. Ed.* 2023, **62**, e202305355), but the proportion of the pathway that generates $\cdot\text{O}_2^-$ intermediates is minor in TPC-3D. Thus, the impact of the reaction between organic matters and $\cdot\text{O}_2^-$ on the system is correspondingly small. To investigate the percentage of ORR pathway involving $\cdot\text{O}_2^-$ intermediates, superoxide dismutase (SOD) was used as a scavenger of $\cdot\text{O}_2^-$ for photocatalytic experiments (Fig. R11). In the presence of SOD, the H₂O₂ yield of TPC-3D decreased by less than one-fifth, confirming that the

percentage of ORR pathway involving $\cdot\text{O}_2^-$ intermediates is minor.

Fig. R12. Investigation of proton source and electron source. **a**, Schematic diagram of proton and electron sources in the ORR reaction of AQ in the presence of phenol. **b**, Reaction energies for the generation of AQH₂ from AQH via four different proton, electron source pathways discussed in Fig. R11a. **c**, The withstand interference ability test. Effect of water matrix components (coexisting anions) and dissolved organic matters (phenol as an example) on the photocatalytic performance of TPC-3D. Reaction conditions were: [photocatalyst] = 0.02 g·L⁻¹, [anions] = 5 mM, [phenol] = 5 ppm.

On the other hand, dissolved organic matters (DOMs) in real water have been reported to provide protons (*J. Am. Chem. Soc.* 2022, **144**, 2603–2613) or act as electron donors (*Appl. Catal. B-Environ.* 2020, **269**, 118770; *Adv. Sustainable Syst.* 2019, **3**, 1900027; *Environ. Sci. Technol.* 2014, **48**, 12679–12688), thereby promoting photocatalytic performance. Thus, we adopted theoretical calculations to elucidate the sources of protons and electrons in this system. The composition of DOMs was complex, and to simplify the model, phenol was used as an example. As illustrated in Fig. R12a, upon light excitation, the electron donor generated an electron and transferred to AQ, where it reacted with the C=O double bond in AQ to form AQH. At this point, an additional electron and a proton were required to form anthrahydroquinone (AQH₂) (*Proc. Natl. Acad. Sci. U. S. A.* 2021, **118**, e2115666118). In the presence of phenol, there were four possibilities for proton and electron sources in this step: H₂O provided protons and electrons, phenol provided protons and electrons, water provided protons while phenol provided electrons, and phenol provided protons while H₂O provided electrons. The reaction energies (ΔE) of these

four pathways were obtained by theoretical calculation as 2.30 eV, 0.89 eV, 3.55 eV and 5.58 eV respectively (Fig. R12b), indicating that AQH preferred proton-coupled electron transfer through phenol rather than H₂O to form AQH₂. Furthermore, phenol was added into pure water for photocatalytic experiments and the photocatalytic performance was improved significantly (Fig. R12c). Thus, these results substantiate that organic pollutants can supply electrons and protons, facilitating the photocatalytic production of hydrogen peroxide.

We have added the discussion in the revised manuscript [pages 8–9].

Q9. I would also suggest a change to the title and abstract. Especially the title (but also the abstract) is very focused on performance so it is not immediately clear what has actually been done.

Response:

Thanks for your suggestion. The title and abstract has been modified in revised manuscript. The revised title and summary are below:

Title: A record-breaking solar-to-chemical conversion efficiency up to 3.6% in ambient conditions through inhibiting interlayer transportation

Abstract: Efficiently converting solar energy into chemical energy remains a formidable challenge in artificial photosynthetic systems. To date, no artificial photosynthetic system operating in the open air has surpassed the highest solar-to-biomass conversion efficiency (1%) observed in plants. In this study, we present a three-dimension polymeric photocatalyst achieving a record-breaking solar-to-H₂O₂ conversion efficiency of 3.6% under ambient conditions, including real water, open air, and room temperature. The excellent performance is attributed to the efficient storage of electrons inside materials via ultrafast intramolecular charge transfer, and the fast extraction the stored electrons by O₂ that can diffuse into the internal pores of the self-supporting three-dimension material. This construction strategy suppresses the interlayer transfer of excitons, polarizers and carriers, effectively increases the utilization of internal excitons to 82%. This breakthrough provides a novel perspective to substantially enhance photocatalytic performance and bear substantial implications for sustainable energy generation and environmental remediation.

Q10. Some parts of the manuscript are over-hyped (see also comment above). For example it is claimed that this is the first instance where SCC has surpassed plants, but just this year has been reported a 7% water splitting efficiency (Nature (<https://www.nature.com/>) 613, pages 66–70 (2023)). Perhaps the authors meant the first instance of a polymeric semiconductor for solar to H₂O₂ production?

Response:

The manuscript mentioned that “the first instance where SCC has surpassed the highest solar-to-biomass conversion (SBC) rate of typical plants in open air”. We apologize for the ambiguity caused by the misplacement of the word "in open air" in the sentence. The correct word order for the sentence should be "the first instance

where SCC in open air has surpassed the highest solar-to-biomass conversion (SBC) rate of typical plants". Here, "in open air" specifically modifies "SCC", indicating the environmental context of the solar-to-chemical conversion being discussed.

The article mentioned by reviewer (*Nature* 2023, **613**, 66–70. Hereinafter referred to as *Nature*) achieves ~7% solar-to-hydrogen (STH) efficiency. While notable, but it does not meet the qualifications we mentioned previous—"SCC in open air". Additionally, the disparity in experimental conditions precludes direct comparison with our system. The differences are as follows:

First, our manuscript emphasize "in open air", i.e., the reaction system is non-closed, completely exposed to air and at normal atmospheric pressure. *Nature* describes the reaction conditions as "The holder was installed on the bottom of the Pyrex chamber, and covered by a vacuum-tight quartz lid and a vacuum-tight plastic ring. Before the photocatalytic reaction, the chamber was vacuumized to a reduced pressure". It follows that its photocatalytic reaction was neither carried out in open system nor under atmospheric pressure, and does not meet the qualification of "in open air".

Second, the temperature was different when testing the SCC efficiency. Our photocatalytic system was carried out at room temperature, with a temperature increase of less than 3°C before and after the reaction. And the *Nature* article claimed that it had designed a temperature-controllable system, and its success in achieving ~7% STH efficiency using tap water or seawater originates from operating at an optimal reaction temperature (~70°C). More notably, *Nature* mentioned that using concentrated solar light without a heat-insulating layer yielded a reaction temperature of only 50.8 °C and an STH efficiency of about 2–3%, which was lower than our SCC (3.6%) in the open system at room temperature.

Third, the light intensity used varies. Our photocatalytic system used a light intensity of 100 mW·cm⁻² (i.e., 1 sun). in contrast, *Nature* specified that the light intensity was 3,800 mW·cm⁻² (i.e., 38 suns), which was an order of magnitude higher than ours, specifically, 38 times higher, yet its SCC efficiency was just less than 2 times higher than ours. Notably, the sunlight energy at the sea surface level is only about 100 mW·cm⁻² (*Nano Energy*, 2023, **108**, 108227), which means a concentrator is required to achieve a light intensity of 3,800 mW·cm⁻² under ambient conditions. And our system achieved a SCC efficiency of 3.6% under ambient conditions.

In summary, much harsher conditions were used in the article (*Nature* 2023, 613, 66–70), so it is not the object of our comparison and does not affect our conclusion that "the first instance where SCC in open air has surpassed the highest solar-to-biomass conversion (SBC) rate of typical plants". In order to make the expression clearer and more explicit, the qualification "in open air" is amended to read "in ambient conditions", i.e. real water, open air and room temperature, and it has been modified in revised manuscript at page 4.

In addition, Table R1 lists some of the excellent SCC performance that can be achieved under natural light conditions (light intensity \leq 100 mW·cm⁻²), and more performance comparisons are available in the Supplementary Table 3. According to our statistics, there are indeed some reports whose SCC performance exceed the

highest solar-to-biomass conversion (SBC) rate of typical plants (1%), but none of them are under ambient conditions (i.e. real water, open air and room temperature). Thus, the reference to “the first instance where SCC in ambient conditions has surpassed the highest solar-to-biomass conversion (SBC) rate of typical plants” in the manuscript is not over-hyped. And the photocatalytic system of this work provides a more feasible way for SCC in ambient conditions.

Table R1. Performance comparison of SCC performance that can be achieved under natural light conditions (light intensity $\leq 100 \text{ mW}\cdot\text{cm}^{-2}$)

Photosynthetic systems	Photocatalysts	Conditions		SCC	Ref.
		Atmosphere	Water body		
H_2O_2	TPC-3D	Air	Pure water	2.4%	This work
		Air	Real water	3.6%	This work
	SA-TCPP	O_2	Pure water	1.20%	Nat. Energy 2023, 8 , 361
	CNIO-GaSA	O_2	Pure water	0.40%	Nat. Synth. 2023, 2 , 557
	Sb-SAPC15	O_2	Pure water	0.61%	Nat. Catal. 2021, 4 , 374
	PM-CDs-30	Air	Real water	0.21%	Nat. Commun. 2021, 12 , 483
	RF523 resin	O_2	Pure water	0.5%	Nat. Mater. 2019, 18 , 985
H_2	C Dots- C_3N_4		Pure water	2.0%	Science. 2015, 347 , 970
	$\text{SrTiO}_3/\text{BiVO}_4$		Pure water	1.1%	Nat. Mater. 2016, 15 , 611
	$\text{SrTiO}_3:\text{Al}$	Closed system	Pure water	0.65%	Nature. 2020, 581 , 411
	HFP		Pure water	1.8%	Angew. Chem. Int. Ed. 2020, 59 , 9653
	CNN/BDCNN		Pure water	1.16%	Nat. Energy 2021, 6 , 388
	$\text{CdTe-4.2/V-In}_2\text{S}_3\text{-3}$		Pure water	1.31%	Nat. Energy 2023, 8 , 504

Q11. Also in the introduction the authors mention that a series of newly develop in situ transient characterisation techniques has been develop. However, in the later discussion, it is not clear what has been actually develop and what the new techniques are.

Response:

In this work, we propose a novel approach for attributing peaks in femtosecond transient absorption spectroscopy, which is challenging and prone to confusion. By utilizing sacrificial agents and small molecule monomers, we differentiate and identify various photoinduced transient species (excitons, polarons, electrons). Based on this approach, the mechanism of accelerated electron extraction from the three-dimensional catalysts on photophysical processes was further explored by switching atmospheres in situ.

We have provided the discussion in the revised manuscript [page 4].

Q12. Since the system reported here by the authors is closely related to other polymer semiconductors for H_2O_2 production, perhaps a discussion on polymer design and the importance of each part is important, also to have a proper comparison to closely-related similar systems.

Response:

The self-supporting three-dimensional (3D) amorphous photocatalyst was synthesized with triptycenes (TPC) as the self-supporting electron donors, built-in redox anthraquinone (AQ) moieties as the electron acceptors, and alkynyl as connectors (Fig. 1a). The three-dimensional structure of triptycenes units exposes more active sites and allows O₂ to diffuse into the interior of the material. Alkynyl bridges have superior electron transfer properties due to the linear conjugated structure. Furthermore, AQ moieties possess a strong electron withdrawal capacity, which promotes the separation of photo-induced carriers. Critically, AQ has a two-electron storage capacity which effectively inhibits the recombination of charge carriers (*Nat. Mater.* 2019, **18**, 985–993; *Nat. Rev. Mater.* 2020, **5**, 828–846). This 3D photocatalyst can rapidly store electrons on AQ by intramolecular transfer and allow O₂ diffuse into the catalyst interior to extract electrons, which suppressing the interlayer transport of photogenerated excitons, polarons, and charge carriers. We have provided the discussion in the revised manuscript [page 5].

And for comparison, the hydrogen peroxide yield of other reported polymer semiconductor photocatalysts had been listed in Table R2 and added in Supplementary Table 1.

Table R2. The comparison of the hydrogen peroxide yield between TPC-3D and other reported polymer semiconductor photocatalysts.

Catalyst	Water body	Atmosphere	Light intensity	H ₂ O ₂ yield / $\mu\text{mol}\cdot\text{g}^{-1}\cdot\text{h}^{-1}$	Ref.
Ag@U-g-C 3N ₄ -NS-1.0	Acidic aqueous solution	O ₂	100 (AM 1.5 G)	~70	Adv. Mater. 2019, 31 , 1806314
Ni4%/O0.2t CN	Water: ethanol=9:1	Air	100 ($\lambda > 400$ nm)	2464	Chem. Eng. J. 2022, 441 , 135999
NiIn ₂ S ₄ -C ₃ N ₄	Water: ethanol= 9:1	O ₂	100 ($\lambda > 400$ nm)	2700	Appl. Catal. B-Environ. 2022, 310 , 121336
g-C ₃ N ₄ /PDI 51	pure water	O ₂	26.9 ($\lambda > 420$ nm)	21	Angew. Chem. Int. Ed. 2014, 53 , 13454– 13459
Co ₁ /AQ/ C ₃ N ₄	pure water	O ₂	100 (AM 1.5 G)	124	Proc. Natl. Acad. Sci. U.S.A. 2020, 117 , 6376
PEI/C ₃ N ₄	pure water	O ₂	100 (AM 1.5G)	208	ACS Catal. 2020, 10 , 3697–3706
g-C ₃ N ₄ /PDI /rGO _{0.05}	pure water	O ₂	43.3 ($\lambda = 420$ -500 nm)	21	J. Am. Chem. Soc. 2016, 138 , 10019– 10025
CTF-BDD BN	pure water	O ₂	44.5 ($\lambda > 420$ nm)	97	Adv. Mater. 2020, 32 , 1904433
RF523	pure water	O ₂	100 (AM 1.5G)	200	Nat. Mater. 2019, 18 , 985–993

RF/P3HT-1	pure water	O ₂	100 (AM 1.5G)	615	J. Am. Chem. Soc. 2021, 143 , 12590– 12599
Sb-SAPC	pure water	O ₂	100 (AM 1.5G)	1179	Nat. Catal. 2021, 4 , 374–384
PQTEE-CO P	pure water	O ₂	$\lambda > 400$ nm	3009	Chem. Eng. J. 2023, 454 , 139929
Bpt-CTF	pure water	O ₂	$\lambda > 400$ nm	3268	Adv. Mater. 2022, 34 , 2110266
AQTEE-C OP	pure water	O ₂	$\lambda > 400$ nm	3204	ACS Catal. 2022, 12 , 12954–12963
ZnPPc-NB CN	pure water	Air	100 ($\lambda > 400$ nm)	114	Proc. Natl. Acad. Sci. U.S.A. 2021, 118 , e2103964118
TPE-AQ	pure water	Air	100 ($\lambda > 400$ nm)	909	Proc. Natl. Acad. Sci. U.S.A. 2021, 118 , e2115666118
TPT-3	pure water	Air	100 ($\lambda > 400$ nm)	1351	Appl. Catal. 2022, 314 , 121488
KDBT	pure water	Air	149	~145	Nat. Commun. 2024, 15 , 1313
	pure water	Air	100 ($\lambda > 400$ nm)	2368	
TPT-alkyny l-AQ	pure water	Air	ambient conditions	2676	Proc. Natl. Acad. Sci. U.S.A. 2022, 119 ,
	river water	Air	ambient conditions	2464	e2202913119
	seawater	Air	ambient conditions	2746	
LBOB	seawater	Air	$\lambda = 427$ nm	497	J. Am. Chem. Soc. 2022, 144 , 2603–2613
	pure water			5940	
TPC-3D	Lake water	Air	100	9991	This work
	River water		($\lambda > 400$ nm)	9615	
	Seawater			9257	

Reviewer #2

This manuscript reports a polymeric photocatalyst achieving a solar-to-H₂O₂ conversion efficiency of 3.6% under ambient conditions. The performance can be attributed to the rapid intramolecular electron transfer and the spontaneous dissociation of excitons. However, the manuscript should be further improved in terms of the following issues before publication.

Response:

Thank you for your insightful feedback on our study. Following your recommendations, we have undertaken targeted clarifications and made substantive editorial enhancements to elevate the overall presentation and clarity of our work.

Comments and Suggestions for Authors:

Q1. On page 9, the authors declared that “DFT calculation revealed that O₂ was dominantly chemisorbed on the alkynyl moieties to form the endoperoxide species in TPC-3D, rather than the AQ or triptycene (TPC) moieties.” Is there any direct evidence that oxygen is chemically adsorbed in the alkynyl group of TPC-3D? Does the alkynyl group’s Raman or FTIR peak change during the photocatalytic reaction as oxygen is stored? Can it be observed by in situ experiments? For example, in situ Raman or in situ FTIR.

Response:

Following your suggestion, in situ diffuse reflectance infrared Fourier transform spectroscopy (DRIFTS) was conducted to explore the adsorption behavior of O₂ on alkynyl moieties. As shown in Fig. R13a, the peak of C≡C bond stretching vibration (2219 cm⁻¹) increased gradually with the injection of O₂ in the dark. The results indicated that alkynyl groups served as the active site for O₂ adsorption as the adsorbed intermediate could result in increased force constants due to symmetry breaking (*Adv. Mater.* 2022, **34**, 2107480). In contrast, under irradiation, the signals of alkynyl moieties decreased, suggesting that alkynyl participates in the photocatalytic ORR (Fig. R13b). We have provided the discussion in the revised manuscript [pages 10–11].

Fig. R13. a–b, In-situ DRIFTS of TPC-3D (a) adsorbing O₂ in the dark and then (b) irradiating.

Q2. On pages 15 and 16, the authors claimed that E_{ar} is the activation energy of charge recombination and the reverse potential barrier, respectively. I wonder if these two identical abbreviations mean the same thing.

Response:

E_a and E_{ar} represent different transfer directions between the lowest singlet excited state and the charge separation state. As shown in Fig. 9c, the value of excitonic binding energy (E_b) is positive and the activation energy (E_a) is from the lowest singlet excited state to the charge separated state to the lowest singlet excited state, in which E_a is the activation energy of exciton dissociation. In Fig. 9d, the exciton binding energy (E_b) is negative, so the barrier is reversed. The reverse potential barrier (E_{ar}) implies a return from the charge separated state to the lowest singlet excited state, which is the activation energy of the charge recombination. Thus, on pages 15 and 16, the activation energy of charge recombination and the reverse potential barrier mean the same thing. To avoid ambiguity, we have changed the expression to the same terminology, i.e. activation energy of charge recombination (E_{ar}), and have modified in revised manuscript at page 19.

Fig. 9. **c**, Illustration of the mutual transitions between the charge separated state (CS) and the lowest singlet excited state (S_1) in PYR-2D. **d**, Illustration of mutual transitions between CS and S_1 in TPC-3D.

Q3. The authors tested the EPR spectra of photocatalysts under ambient conditions in light and darkness, thus claimed that the results show that AQ can store electrons. Because electrons stored in the structure can undergo recombination or ORR involving oxygen and water in the dark, the EPR tests described in the paper does not appear to represent the electronic signal during the real reaction process. It is recommended to test the EPR spectra of the material under reaction conditions. In addition, what happened to the EPR signal once the trapped electrons finished the reaction in the dark?

Response:

Following your suggestion, the time-dependent in-situ EPR spectra measurements have been conducted under reaction conditions with the following results:

As depicted in the Fig. R14a, the single Lorentzian line of TPC-3D was detected in H₂O with the center g factor of 2.0038, which derived from the unpaired electrons in the oxygen-centered by the storage of one electron in AQ(Angew. Chem. Int. Ed. 2023, 62, e202218115). This signal was gradually enhanced with the increasing light duration which was due to electrons generated by photoexcitation were stored on AQ to form a single-electron stored AQH through intramolecular transfer, and reached saturation at about 10 min (Fig. R14b).

Fig. R14. Detection of the unpaired electrons in the oxygen-centered active sites. **a**, The time-dependent in-situ EPR spectra of TPC-3D in H₂O. **b**, Fitted curve of the highest point of the EPR spectrum in H₂O ($g = 2.0058$) versus time. **c**, The time-dependent in-situ EPR spectra of TPC-3D in air. **d**, Key steps of H₂O₂ production by two-electron ORR process on AQ. **e**, The time-dependent in-situ EPR spectra of TPC-3D in SafeDry acetonitrile. **f**, Fitted curve of the highest point of the EPR spectrum in SafeDry acetonitrile ($g = 2.0058$) versus time.

Later, turned off the light and the EPR signal decreased rapidly over a 10-minute period, after which it progressively stabilized (Fig. R14a, b). There are three possible mechanisms that could lead to the signal decrease after the light is turned off: the single electrons on AQH recombine, react with oxygen, or react with H₂O. To investigate the cause of the signal reduction, the solid was directly exposed to air for EPR measurement (Fig. R14c). The results showed that signal intensity had no significant change before and after the lights were turned off for 1 h, indicating that the single-electron stored on AQH neither recombine nor directly react with O₂ in air. Therefore, the decline in the signal can be attributed to the reaction of electrons with H₂O. Specifically, the single-electron stored AQH gains an additional electron from H₂O to form the two-electron stored AQH₂ which subsequently reacted with O₂ to release H₂O₂ (Fig. R14d), leading to the signal decrease (*Nat. Commun.* 2024, 15, 2649). In order to demonstrate that AQH receives electrons from H₂O, EPR measurement was conducted in SafeDry acetonitrile to eliminate the interference of H₂O. As shown in Fig. R14e, f, there was no significant change in the signal under both light irradiation and light off conditions, confirming that H₂O is the crucial electron donor.

We have provided the discussion in the revised manuscript [page 12] and Supplementary Note 4, and additional data have been added to the Supplementary Fig. 34.

Q4. Please check the legend carefully for mistakes. For example, TCL-2D was incorrectly spelled as TCL-3D in supplementary 19a.

Response:

We thank the reviewer for pointing out the error. The incorrectly written legend has been corrected.

Q5. The authors claimed that the electrons stored under dark conditions come from electron-hole pairs produced by light excitation. Since electrons and holes are formed in pairs, what process do the holes in TPC-3D go through during the reaction (from light to darkness)?

Response:

Under light irradiation, the excited TPC-3D generates photo-induced holes that can participate in either a four-electron water oxidation reaction (WOR) to produce O₂ or a two-electron WOR to yield H₂O₂. (*Adv. Mater.* 2020, 32, 1904433). Rotating ring-disk electrode (RRDE) measurement confirmed the existence of both pathways in TPD-3D (Fig. R15a, b). Moreover, under bubbling argon to remove O₂ from the air, with the incorporation of electron sacrificial agents (AgNO₃) to inhibit the generation of H₂O₂ from ORR, the formation of H₂O₂ could also be observed (Fig. R15c), along with the H₂¹⁸O isotope experiments (Fig. R15d), demonstrating that target product H₂O₂ can be generated via WOR pathway. The time-dependent DFT (TD-DFT) calculations indicated that all the active sites were mainly located on the TPC and alkynyl moieties for WOR, as the holes were primarily occupied in these two sites in

the excited states (Fig. R15e).

Fig. R15. **a**, The potential of Pt ring electrodes was set at +0.6 V versus Ag/AgCl to detect H₂O₂. **b**, The potential of Pt ring electrodes was set at -0.23 V versus Ag/AgCl to detect O₂. **c**, H₂O₂ production through WOR. The amount of H₂O₂ production under the conditions of bubbling argon gas to exclude O₂ from the air and adding electron sacrificial agents to inhibit the generation of H₂O₂ from oxygen reduction. **d**, H₂¹⁸O isotope experiment to explore the H₂O₂ evolution through WOR pathway for TPC-3D. Before the addition of MnO₂, there were weak ¹⁸O₂ and ¹⁶O₂ signals in the system were weak, which was attributed to trace of O₂ from the reaction environment. When isotopically labeled water (H₂¹⁸O) was employed in the photocatalytic reaction instead of conventional water (H₂¹⁶O), and the subsequently generated hydrogen peroxide was converted to O₂ by MnO₂, a distinct enhancement of the isotopic signal peak corresponding to ¹⁸O₂ was observed. **e**, The analysis for the distribution of the holes (blue) and electrons (green) for TPC-3D by Time-dependent DFT (TD-DFT) calculation.

After removing the light source, the remaining holes are less susceptible to recombine with electrons which have been stored. Moreover, due to the absence of hole storage sites, it cannot be stabilized and can only be consumed by reacting with H₂O or by engaging in alternative annihilation pathways. Afterwards, new holes are no longer produced due to the lack of external light excitation.

We have provided the discussion in Supplementary Note 4.

Q6. The specific surface area is analyzed through the BET, however, the catalytic active area is not equal to the specific surface area. Thus, the analysis of real catalytic active area should be added.

Response:

The oxygen reduction reaction (ORR) primarily takes place at two distinct sites on TPC-3D: the alkynyl group (site I) and the AQ (site II). As indicated by DFT calculations (Fig. R16a) and in-situ DRIFTS experiments (Fig. R16b), site I acted as a

catalytically active site for ORR upon the adsorption of oxygen, and then produced H_2O_2 . The O_2 temperature programmed desorption (O_2 -TPD) experiment depicted in Fig. R16c reveals that TPC-3D had a significantly higher chemisorption capacity for oxygen compared to PYR-2D and TPL-2D, implying a greater abundance of active alkynyl sites.

For site II, TPC-3D displayed pronounced EPR peaks at $g=2.0038$ under light irradiation (Fig. R16d), attributed to the oxygen-centered radicals resulting from the storage of one electron in AQ (*Angew. Chem. Int. Ed.* 2023, **62**, e202218115). TPC-3D exhibited the strongest radical signals among the three conjugated polymers (CPs), indicating a higher number of active AQ sites. In addition, TPC-3D demonstrated the highest water vapor adsorption capacity (Fig. R16e), benefiting WOR.

In conclusion, TPC-3D exhibited more active sites for ORR and WOR than the photocatalysts with 2D structure. In addition, BET experiments (Supplementary Fig. 28) showed that TPC-3D had a larger specific surface area due to its three-dimensional structure, allowing better exposure of the active sites. The relevant statement has been added on pages 13–14 of the revised manuscript and Supplementary Note 6.

Fig. R16. **a**, Molecular model and adsorption energy (E_{adsorb}) for TPC-3D when oxygen is adsorbed at alkynyl moieties. **b**, In-situ DRIFTS of TPC-3D adsorbing O_2 in the dark and then irradiating. **c**, O_2 -TPD spectra of the CPs. **d**, Electron paramagnetic resonance (EPR) spectra of CPs under illumination for 30 min. **e**, Water vapor adsorption-desorption isotherms of CPs.

Q7. The real water, such as lake water, river water, and seawater, is used for photocatalytic reaction. This is very interesting, but, the element contents of these real water is not mentioned. And how the extra elements in real water affect the photocatalytic performance?

Response:

Supplementary Table 3 | Specific parameters of water samples

	Pure water	Lake water	River water	Seawater
Dissolved oxygen (mg/L)	7.09	8.11	6.51	8.49
pH	6.13	7.53	6.52	8.39
Total Carbon (mg/L)	/	22.21	15.42	26.61
Inorganic carbon (mg/L)	/	14.60	8.49	24.97
Non-purgeable organic carbon (mg/L)	/	7.50	5.95	2.24
Total Nitrogen (mg/L)	/	2.13	1.00	0.53
Ion concentration (mM)	Cl ⁻	/	3.9430	0.2152
	SO ₄ ²⁻	/	0.5680	0.0448
	NO ₃ ⁻	/	0.0008	0.0016
	Br ⁻	/	0.0045	0.0003

Ion concentration in real water had been listed in Supplementary Table 4, but there does not appear to be a direct correlation between performance and Cl⁻, SO₄²⁻, NO₃⁻, and Br⁻ concentrations. In order to investigate the effects of extra elements, photocatalytic experiments were carried out by adding 5 mM of Cl⁻, SO₄²⁻, NO₃⁻, Br⁻, and HCO₃⁻ to pure water. As shown in Fig. R17, Cl⁻, SO₄²⁻, NO₃⁻, and Br⁻ had no significant effect on the photocatalytic performance, indicating that TPC-3D demonstrated exceptional resistance to interference. However, the addition of HCO₃⁻ led to an improvement in photocatalytic performance. This improvement was attributed to HCO₃⁻ acting as a hole scavenger, suppressing the hole-mediated H₂O₂ oxidation ($\text{H}_2\text{O}_2 + 2\text{h}_{\text{VB}}^+ \rightarrow \text{O}_2 + 2\text{H}^+$), and facilitate the generation of electrons (*Chem. Eng. J.* 2022, **432**, 134401). The discussion has been added in the revised manuscript at page 9 and in Supplementary Note 3, and additional data have been provided to the Supplementary Fig. 22.

Fig. R17. The withstand interference ability test. Effect of water matrix components (coexisting anions) and dissolved organic matters (phenol as an example) on the photocatalytic performance of TPC-3D. Reaction conditions were: [photocatalyst] = 0.02 g·L⁻¹, [anions] = 5 mM, [phenol] = 5 ppm.

Q8. During the catalytic reaction, the analysis of adsorption-conversion process seems missing, and this should be discussed by DFT or in situ technique.

Response:

For WOR, the excited TPC-3D localized the photo-induced hole at the carbon of electron donor and alkynyl unit (Fig. R18a). Specifically, for TPC-3D, WOR occurred at carbon 3 (on triptycenes) and carbon 37 (on alkynyl group) (Fig. R18b). The calculation results indicated a preference for the adsorption of OH* intermediates on alkynyl (Fig. R18c), enabling H₂O to form the target product H₂O₂ through a two-electron WOR. The same phenomenon was observed in PYR-2D and TPL-3D. Additionally, the in situ diffuse reflectance infrared Fourier transform spectroscopy (DRIFTS) under H₂O conditions displayed relatively obvious infrared absorption peaks at 2219, 1573, and 1072 cm⁻¹, which were attributed to C≡C, benzene ring, and C-OH absorption, respectively (Fig. R18d) (*Adv. Mater.* 2022, **34**, 2107480; *Angew. Chem. Int. Ed.* 2023, **62**, e202309480; *Angew. Chem. Int. Ed.* 2023, **62**, e202218318). And new peaks emerged around 1184–1203 cm⁻¹, corresponding to the infrared absorption of -OH from generated H₂O₂. These results confirmed the active participation of alkynyl and triptycenes structures in the production of H₂O₂ through the WOR. Isotope experiments also provided strong evidence for WOR (Fig. R18e).

Fig. R18. **a**, The analysis for the distribution of the holes (blue) and electrons (green) for CPs by Time-dependent DFT (TD-DFT) calculation. **b**, The contribution of non-hydrogen atoms to holes and electrons in excited state and the corresponding atom labels. **c**, Calculated free energy diagrams of two-electron water oxidation pathways toward H₂O₂ production on acetylene and electron-donor sites in TPC-3D. **d**, In-situ DRIFTS under H₂O of TPC-3D. **e**, H₂¹⁸O isotope experiment to explore the H₂O₂ evolution through WOR pathway for TPC-3D.

On the other hand, for ORR, DFT calculations revealed that O₂ could spontaneously adsorb on the alkynyl moieties to form the endoperoxide species in TPC-3D (Fig. R19a). To further explore the adsorption behavior of O₂ on alkynyl moieties, in situ diffuse reflectance infrared Fourier transform spectroscopy (DRIFTS) was conducted. As shown in Fig. R19b, the stretching vibration of the C≡C bond

(2219 cm^{-1}) increased gradually with the injection of O_2 in the dark. The results indicated that alkynyl groups served as the active site for O_2 adsorption as the adsorbed intermediate could result in increased force constants due to symmetry breaking (*Adv. Mater.* 2022, **34**, 2107480). In contrast, under irradiation, the signals of alkynyl moieties decrease, suggesting that alkynyl participates in the photocatalytic ORR (Fig. R19b). This portion of O_2 adsorbed on the alkynyl moieties was proven capable of directly reacting with e^- and being converted to H_2O_2 , since H_2O_2 could still be successfully produced under the condition of in Ar atmosphere to exclude O_2 from the air and adding hole sacrificial agents added to inhibit the generation of O_2 from water oxidation (Fig. R19c).

Fig. R19. **a**, Molecular model and adsorption energy (E_{adsorb}) for TPC-3D when oxygen is adsorbed at alkynyl moieties. **b**, In-situ DRIFTS of TPC-3D adsorbing O_2 in the dark and then irradiating. **c**, The amount of H_2O_2 production under the conditions that oxygen was injected first, then bubbled argon gas to exclude O_2 from the air and adding hole sacrificial agents to inhibit the generation of O_2 from water oxidation. **d**, Key steps of H_2O_2 production by two-electron ORR process on AQ. **e**, In-situ DRIFTS of light in argon and dark in oxygen process of TPC-3D.

Meanwhile, another oxygen reduction pathway that reduced AQ reacted with O_2 was also proposed. The TD-DFT calculations showed that the active sites were mainly located on the carbonyl oxygen (atoms 35 and 36 in Fig. R18b) and carbonyl carbon (atoms 27 and 28 in Fig. R18b) of AQ moieties for ORR. As shown in Fig. R19d, under visible-light irradiation, the electron donor generated an electron and transferred to AQ, where it reacted with the $\text{C}=\text{O}$ double bond to form $\text{C}-\text{O}$ oxygen-centered organic radicals, and combined with proton released from the H_2O molecule initiated an electron-coupled hydrogenation of AQ to form AQH. Subsequently, H_2O provided another proton and an electron, resulting in resulting in

the formation of a two-electron storage of anthrahydroquinone (AQH₂) (*Proc. Natl. Acad. Sci. U. S. A.* 2021, **118**, e2115666118; *ACS Catal.* 2022, **12**, 12954–12963). AQH₂ intermediate was oxidized by the adsorbed oxygen molecules to generate AQH₂-1,4-endoperoxide by electron transfer. Afterwards, 1,4-endoperoxide species coupled the adjacent hydrogen in the hydroxyl group of H₂AQ to release H₂O₂ and regenerated the AQ redox center (*Nat. Commun.* 2024, **15**, 2649). In DRIFTS as shown in Fig. R19e demonstrated the transition from AQ (1680 cm⁻¹) to AQH (1358 cm⁻¹), AQH₂ (1482 cm⁻¹) under irradiation (*Proc. Natl. Acad. Sci. U. S. A.* 2021, **118**, e2115666118). Furthermore, with the injection of O₂, a significantly enhanced infrared vibration peak located at 910 cm⁻¹ could be observed which was assigned to the 1,4-endoperoxide intermediate species by O₂ adsorption on the carbon of AQH₂ (*Angew. Chem. Int. Ed.* 2023, **62**, e202305355), further corroborating the ORR on AQ. We have provided the discussion in the revised manuscript [pages 10–12].

Q9. During the catalytic reaction, did the amorphous carbon make contribution to the performance? How?

Response:

In this work, all the photocatalysts employed are amorphous. Amorphous polymers boast a wealth of advantages due to their rich and tunable structures, along with the mild synthesis conditions. In contrast, highly crystalline covalent organic frameworks (COFs) require the use of reversible covalent bonds, which are easily decomposable, and their stringent synthesis conditions significantly limit their application in practical production (*Nature* **2022**, 604, 72–79).

These materials as a whole exhibit amorphous properties, but differences in microenvironments impart polarity to each component. They are bifunctional photocatalysts assembled with alternating electron donors and acceptors composed of carbon atoms, which promote the rapid separation and migration of photogenerated electrons and holes for efficient redox reactions.

Specifically, the strategy of alternately linking triptycene with strong electron-donating capability and AQ with strong electron-accepting capability enabled the spontaneous separation of excitons in the excited TPC-3D (Fig. R20a). TDDFT analysis showed that the holes and electrons fully dissociated and distributed across completely different sites (Fig. R20b). In detail, the holes were located at carbon 3 (on triptycene) and carbon 37 (on the acetylene unit), as shown in (Fig. R20c), indicating that the water oxidation reaction occurred at these two sites. OH^{*} intermediates were adsorbed on the active sites to form the target product H₂O₂ through two-electron WOR. Meanwhile, electrons are located on the carbonyl oxygen (atoms 35 and 36) and carbonyl carbon (atoms 27 and 28) of the AQ group (Fig. R20b), pointing to an oxygen reduction reaction at this site. Under visible-light irradiation, AQ sequentially formed two-electron-stored AQH₂ via an electron-coupled hydrogenation reaction. Then, adsorption of oxygen was occurred on AQH₂ and generated H₂O₂. The oxidation and reduction reactions proceed at different sites, allowing the material to efficiently generate H₂O₂.

We have provided the discussion in Supplementary Note 7.

Fig. R20. **a**, Temperature-dependent photoluminescence (TD-PL) of TPC-3D at different temperatures. PL intensity increased with temperature for TPC-3D, indicating the excitons separated spontaneously (*Angew. Chem. Int. Ed.* 2021, **60**, 15348–1535). **b**, The analysis for the distribution of the holes (blue) and electrons (green) for TPC-3D by Time-dependent DFT (TD-DFT) calculation. **c**, The contribution of non-hydrogen atoms to holes and electrons in excited state and the corresponding atom labels.

Q10. In Fig. S7 and Fig. S16, complete XRD data of 3-10 degrees should be provided.

Response:

Thank you for your suggestions. The complete XRD data of 3-80 degrees had been provided in Supplementary Fig. 7 and 16.

Supplementary Fig. 7 | Powder X-ray diffraction (PXRD) patterns of CPs. The PXRD profiles indicate that all the three CPs exhibited the features of amorphous carbon.

Supplementary Fig. 16 | Post-characterization of TPC-3D after photocatalysis. a, PXRD image of the as-prepared TPC-3D and TPC-3D after 5 cycles of photoreaction.

Q11. There are some support information graphs that are not mentioned in the article, such as S20 and S43. Please check carefully.

Response:

Thanks to the reviewer for pointing out our omissions. In the revised manuscript, all figures in supplementary information have been mentioned in the article.

Q12. How is the stability of TPC-3D in pure water and real water (including lake water, river water, and sea water)? Is it only limited to 5 hours or 5 cycles?

Response:

To explore the stability of TPC-3D, the material was dispersed in pure water for five additional cycles, and the hydrogen peroxide yield maintained at $\sim 4500 \mu\text{mol}\cdot\text{g}^{-1}\cdot\text{h}^{-1}$ (Fig. R21a). Considering the laborious and time-intensive process of dispersing the catalyst in water and subsequent collection for the next cycle, along with substantial material loss during these operations, we immobilized TPC-3D on a glass slide using Nafion as an adhesive to construct the device shown in Fig. R21b. Once the device was assembled, the materials could be used repeatedly without the need for collection. And then 15 cycles of photocatalysis were performed in real water employing the device and remained well-stabilized for 15 cycles (Fig. R21c). We have provided the discussion in the revised manuscript [page 7].

Fig. R21. a, The 5th to 10th cycle experiments of TPC-3D in pure water. b, Immobilization of TPC-3D powders (2 mg) onto a glass slide (20 mm × 20 mm) for H₂O₂ production in 400 mL of lake water under xenon-lamp irradiation ($\lambda > 400 \text{ nm}$, light intensity: $100 \text{ mW}\cdot\text{cm}^{-2}$). c, The stable H₂O₂ production activity of TPC-3D under the condition described in Fig. R23b.

REVIEWERS' COMMENTS

Reviewer #1 (Remarks to the Author):

The authors have addressed the concerns and the issues. The manuscript can now be accepted.

Reviewer #2 (Remarks to the Author):

The author answered the questions well now.